# Investigating Pattern Neurons in Urban Time Series Forecasting

**Chengxin Wang   Yiran Zhao   Shaofeng Cai   Gary Tan**
National University of Singapore
{cwang,yiran,shaofeng,gtan}@comp.nus.edu.sg

## ABSTRACT

Urban time series forecasting is crucial for smart city development and is key to sustainable urban management. Although urban time series models (UTSMs) are effective in general forecasting, they often overlook low-frequency events, such as holidays and extreme weather, leading to degraded performance in practical applications. In this paper, we first investigate how UTSMs handle these infrequent patterns from a neural perspective. Based on our findings, we propose **P**attern **N**euron guided **Train**ing (`PN-Train`), a novel training method that features (i) a *perturbation-based detector* to identify neurons responsible for low-frequency patterns in UTSMs, and (ii) a *fine-tuning mechanism* that enhances these neurons without compromising representation learning on high-frequency patterns. Empirical results demonstrate that `PN-Train` considerably improves forecasting accuracy for low-frequency events while maintaining high performance for high-frequency events. The code is available at `https://github.com/cwang-nus/PN-Train`.

## 1 INTRODUCTION

Recent advancements in urban time series models (UTSMs) have significantly improved forecasting accuracy, facilitating smart city applications such as optimizing metropolitan transit, managing pedestrian flow, and enhancing resource allocation for ride-hailing services (Yao et al., 2018; Wu et al., 2020; Ji et al., 2022). While deep learning models (Geng et al., 2019; Jiang et al., 2023; Gao et al., 2023) have shown great promise in urban time series forecasting, existing models focus on capturing cross-variable and temporal dependencies to enhance overall accuracy. However, their performance degrades in many real-world scenarios, especially when forecasting low-frequency events such as extreme weather, emergencies, holidays (Lee et al., 2022; Lee & Ko, 2024). Accurate forecasting of these events is crucial for resource management, allowing ride-hailing companies to adjust fleets and transit systems to modify schedules, so as to optimize operations and reduce costs during fluctuating demand (Zhang et al., 2017; Park et al., 2020).

Urban time series data exhibits distinct patterns for high- and low-frequency events (Lee et al., 2019; Wang et al., 2019). As the example illustrated in Figure 1, while patterns within each category remain consistent, significant differences exist between holiday, weekday, and weekend patterns. Specifically, weekdays and weekends represent high-frequency patterns, occurring regularly throughout the year, whereas holidays are low-frequency, spanning fewer than 15 days, or approximately 4% of the year. Deep learning models often struggle to predict low-frequency events, such as holidays in the example, largely due to their bias toward majority patterns and the scarcity of data for these rare occurrences. Several studies have attempted to improve holiday forecasting, e.g., using exponential-growth models (Wang et al., 2019)

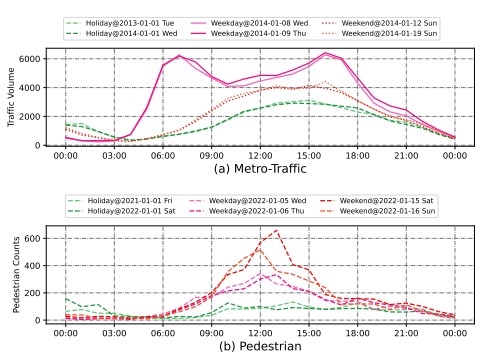

Figure 1: Examples of high-frequency patterns during weekdays and weekends, contrasted with low-frequency patterns on holidays.

and support vector regression (Luo et al., 2019a). More recently, deep learning models have introduced memory architectures for retrieving patterns from a pattern bank (Lee et al., 2022; Li et al., 2022) and dynamic positional embeddings to implicitly capture various patterns (Shao et al., 2022; Liu et al., 2023), leading to enhanced forecasting accuracy. However, despite improved forecasting accuracy, understanding the underlying mechanisms through which these models capture low-frequency patterns at the neuron level remains unexplored.

In this paper, inspired by the *knowledge neurons* in large language models (LLMs) (Dai et al., 2022; Zhao et al., 2024), we investigate two fundamental questions: (1) Do neurons associated with low-frequency patterns exist in UTSMs? (2) If so, how can we enhance the representation learning of these neurons to improve urban time series forecasting?

To answer these questions, we perform an in-depth analysis of UTSMs at the neuron level. First, we introduce a Pattern Neuron Detector (PND), which identifies *pattern neurons*, i.e., neurons strongly correlated with low-frequency patterns, using a perturbation-based approach. This method evaluates neuron importance by measuring the impact of perturbations on the model's output features. Next, we employ a Pattern Neuron Verifier (PNV) to quantify how these neurons impact forecasting performance by deactivating them, so as to confirm that neurons specifically tied to certain patterns indeed exist in UTSMs. Based on our findings, we propose **P**attern **N**euron guided **Train**ing (`PN-Train`), a novel training method that detects these pattern neurons and fine-tunes them using a Pattern Neuron Optimizer (PNO) to improve forecasting for low-frequency patterns while maintaining performance for high-frequency patterns. We summarize our main contributions as follows:

- We conduct the first investigation into neurons associated with low-frequency patterns in urban time series models (UTSMs) and confirm their existence.
- We introduce `PN-Train`, a pattern neuron-guided training method for urban time series forecasting, which effectively detects these neurons using a perturbation-based detector.
- We propose a fine-tuning mechanism that enhances the representation learning of detected pattern neurons, significantly improving forecasting accuracy.
- Extensive experiments demonstrate that `PN-Train` significantly improves the forecasting accuracy of state-of-the-art methods across real-world datasets.

## 2 PRELIMINARIES

**Urban Time Series Forecasting (UTSF)**  UTSF aims to forecast future time series data using sensor readings collected from urban environments. The objective is to predict $H$ future values $\mathbf{x}_{\tau:\tau+H}$ at each time step $\tau$, using a learnable model, UTSM, which leverages a look-back window of $L$ past observations $\mathbf{x}_{\tau-L:\tau}$. Additionally, auxiliary features $E$, such as the time of day, day of the week, holiday indicators, etc., are incorporated to enhance the forecasting process. Formally, the prediction task can be formulated as $\hat{\mathbf{x}}_{\tau:\tau+H} = \text{UTSM}(\mathbf{x}_{\tau-L:\tau}, E)$. To ensure accurate forecasting, the UTSM is typically trained using the Mean Absolute Error (MAE) loss function, defined as $\mathcal{L} = \frac{1}{H} \sum_{h=1}^{H} \|\hat{\mathbf{x}}_{\tau+h} - \mathbf{x}_{\tau+h}\|_1$.

**Pattern Neurons**  In a neural network, individual neurons contribute differently to the representation and memorization of various patterns. Let $I(h_i, p)$ denote the influence of neuron $h_i$ on pattern $p$. A *pattern neuron* can then be defined as a neuron with a strong influence on a specific pattern, namely $I(h_i, p)$ is high for the particular pattern $p$.

**Low-Frequency Event & Pattern**  In urban time series data, certain events, such as holidays or extreme weather, occur infrequently and exhibit a low occurrence rate (Devore, 2000). These events are defined as *low-frequency events* as their occurrence rate, $S/Z$, falls below a small threshold, where $S$ is the total number of event occurrences and $Z$ is the total number of observation days. The *low-frequency pattern* is the distinctive data behavior of these events.

## 3 PN-TRAIN

In this section, we introduce `PN-Train`, a training method designed to enhance urban time series forecasting for low-frequency patterns by identifying and fine-tuning the pattern neurons associated with these patterns in the UTSM. The overall architecture of `PN-Train` is illustrated in Figure 2.

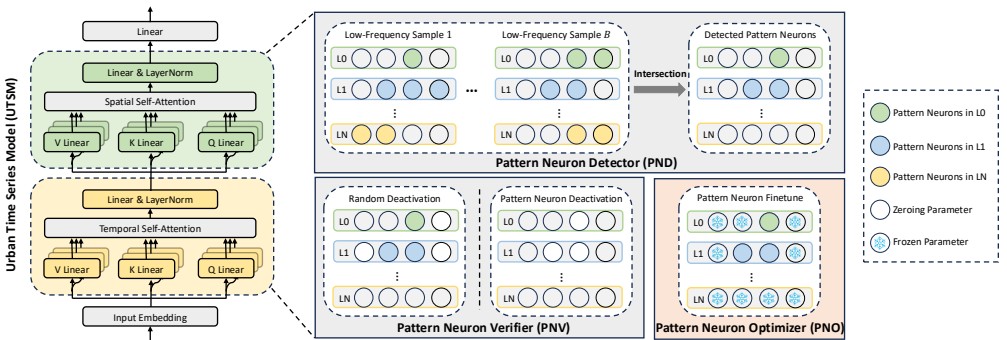

Figure 2: The architecture of PN-Train, which consists of four components: Urban Time Series Model (UTSM) captures time series patterns from historical data; Pattern Neuron Detector (PND) identifies neurons associated with specific patterns, such as low-frequency samples; Pattern Neuron Verifier (PNV) validates the detected neurons; and Pattern Neuron Optimizer (PNO) fine-tunes the UTSM at the neuron level. LX represents the X-th linear layer in the UTSM.

## 3.1 PATTERN NEURON DETECTOR FOR URBAN TIME SERIES MODELS

Typically, Urban Time Series Models (UTSMs) are designed to capture patterns from historical data (Zhou et al., 2020; Liu et al., 2020; 2022). While they can effectively learn frequent patterns thanks to sufficient training data, the distribution of patterns is often imbalanced in practice (Luo et al., 2019a; Lee & Ko, 2024). In particular, high-frequency events like weekdays and weekends are well-represented, making them easier to learn. In contrast, low-frequency events, such as holidays, have fewer training samples, which results in reduced forecasting performance (Krawczyk, 2016; Smyl et al., 2023). We hypothesize that some neurons in UTSMs are already tuned to capture low-frequency patterns based on past encounters with these events. To test this hypothesis, we introduce a Pattern Neuron Detector (PND) to identify neurons linked to low-frequency patterns.

**Neurons in UTSMs**  UTSMs nowadays are effective at learning patterns from historical data, and a key component of these models is the linear layer, which is central to pattern learning and memorization (Geva et al., 2021; Dai et al., 2022). In this work, we focus on a transformer-based UTSM (Liu et al., 2023), which employs both linear layers and self-attention layers to model temporal correlations (patterns over time) and spatial correlations (relationships across urban locations). The detection of pattern neurons in the UTSM is carried out using our proposed Pattern Neuron Detector (PND), which can be applied to both linear layers and self-attention layers as introduced below.

**Pattern Neuron Detector (PND)**  Inspired by *Knowledge Neurons* (Tang et al., 2024; Zhao et al., 2024) in large language models (LLMs), which identify neurons with high activation values, we define *pattern neurons* in UTSMs whose contributions to forecasting targets are significant in the perturbation assessment. Specifically, the influence $h_i^k$ of the $k$-th neuron at the $i$-th layer can be quantified by comparing the model outputs when the neuron is deactivated:

$$I(h_i^k|\mathbf{x}^p) = \left\| UTSM(\mathbf{x}^p, \boldsymbol{W}) - UTSM(\mathbf{x}^p, \boldsymbol{W}\backslash\boldsymbol{w}_i^k) \right\|_1, \tag{1}$$

where $\mathbf{x}^p$ represents an input with the pattern $p$, $\boldsymbol{W}$ and $\boldsymbol{w}_i^k$ denote the weights of the UTSM and the weights of the neurons respectively, and $UTSM(\mathbf{x}^p, \boldsymbol{W}\backslash\boldsymbol{w}_i^k)$ represents the model output with only neuron $h_i^k$ deactivated.

As UTSMs contain a vast number of neurons, deactivating each neuron individually is impractical. Prior work (Zhao et al., 2024) has shown that neurons linked to specific patterns often exhibit high feature activation values. This suggests that neuron activation values can serve as strong indicators of their importance in capturing corresponding patterns. We therefore devise an attribute score $\text{Attr}_p$ to quantify the influence of the $k$-th neuron for a specific pattern $p$ given an input $\mathbf{x}^p$ with this pattern:

$$\text{Attr}_p(h_i^k \mid \mathbf{x}^p) = \left\| \sum_{s,t} f(\mathbf{x}^p, \boldsymbol{w}_i^k)_{s,t,:} \right\|_1, \tag{2}$$

where $s$ and $t$ represent spatial and temporal dimensions respectively, and $f(\mathbf{x}^p, \boldsymbol{w}_i^k)$ is the function to generate the activation values for the $k$-th neuron at the $i$-th layer.

---

**Algorithm 1:** Pattern Neuron Guided Training Method

---

**Input:** The urban time series model $UTSM$; the training dataset $\mathcal{D}_{\text{train}}$ and validation dataset $\mathcal{D}_{\text{val}}$; the size of the detection sample $B$, and the size of the fine-tuning sample $R$; and the learning rates for training, $\alpha_1$, and fine-tuning, $\alpha_2$.
**Output:** The fine-tuned urban time series model $UTSM$

---

   // Process fine-tuning samples and training samples
1  $\mathcal{D}_{\text{finetune}} \leftarrow \text{RandomSample}(\{\mathbf{x} \in \mathcal{D}_{\text{train}} \mid \mathbf{x} \text{ is a low-frequency sample}\}, R)$
2  $\mathcal{D}_{\text{train'}} \leftarrow \mathcal{D}_{\text{train}} \setminus \mathcal{D}_{\text{finetune}}$
   // Train the urban time series model
3  **repeat**
4      Randomly select a batch of instances $\mathcal{S}$ from $\mathcal{D}_{\text{train'}}$
5      Optimize $UTSM$ using AdamW with a learning rate of $\alpha_1$ on batch $\mathcal{S}$.
6  **until** *met the stopping criteria*;
   // Select detection samples and detect the pattern neurons
7  $\mathcal{D}_{\text{detect}} \leftarrow \text{RandomSample}(\{\mathbf{x} \in \mathcal{D}_{\text{train}} \mid \mathbf{x} \text{ is a low-frequency sample}\}, B)$
8  $\mathcal{N}^{p_l} \leftarrow \text{PND}(UTSM, \mathcal{D}_{\text{detect}})$
   // Fine-tune the detected pattern neurons
9  $\hat{\mathbf{y}} \leftarrow UTSM(\mathcal{D}_{\text{finetune}}, \mathcal{N}^p)$
10  $\mathcal{L} \leftarrow \text{MAE}(\hat{\mathbf{y}}, \mathbf{y})$
11  Optimize pattern neurons $\mathcal{N}^p$ using AdamW with a learning rate $\alpha_2$.
   // Return the fine-tuned UTSM
12  **return** $UTSM$

---

To detect pattern neurons, we focus on samples that exhibit the patterns of interest. Specifically, for identifying pattern neurons associated with low-frequency patterns $p_l$, e.g., holidays, we use a set of samples $\{\mathbf{x}_1, \mathbf{x}_2, \ldots, \mathbf{x}_B\}$, where $B$ is the number of samples used for detection, and define pattern neurons as neurons whose attribute scores are high across *all* the $B$ samples:

$$\mathcal{N}^{p_l} = \bigcap_{b=1}^{B} \left\{ n_i^k \mid \text{rank}(\text{Attr}_p(h_i^k \mid \mathbf{x}_b^{p_l})) \leq \epsilon N, \quad \forall i, k \right\} \tag{3}$$

where $\text{rank}(\cdot)$ gives the rank of the attribution score in descending order for the $k$-th neuron at the $i$-th layer, $\epsilon$ is a predefined threshold that determines the fraction of candidate pattern neurons among all the $N$ neurons in the UTSM given a sample $\mathbf{x}_b^{p_l}$.

Notably, such a detection process can be easily applied to self-attention layers, where the query $Q$, key $K$, and value $V$ are the weights of the attention function:

$$\text{Attention}(\mathbf{x}) = \text{softmax}\left(Q(\mathbf{x})K(\mathbf{x})^\top / \sqrt{d}\right) V(\mathbf{x}),$$
$$Q(\mathbf{x}) = f(\mathbf{x}, W_Q), \quad K(\mathbf{x}) = f(\mathbf{x}, W_K), \quad V(\mathbf{x}) = f(\mathbf{x}, W_V). \tag{4}$$

In particular, the attribution scores for these layers can be obtained via Equation 2 using $f(\mathbf{x}, W_Q)$, $f(\mathbf{x}, W_K)$, and $f(\mathbf{x}, W_V)$ respectively as the function $f(\mathbf{x}^p, \mathbf{w}_i^k)$.

### 3.2   Pattern Neuron Verification and Optimization

In this section, we answer the two key research questions: (1) Do pattern neurons exist for low-frequency patterns? (2) Can optimizing these pattern neurons improve the performance of UTSMs? To answer these, we employ a Pattern Neuron Verifier (PNV) to validate the existence of pattern neurons and devise a Pattern Neuron Optimizer (PNO) to enhance UTSM performance by fine-tuning the detected pattern neurons.

**Pattern Neuron Verifier (PNV)**   To validate the existence of pattern neurons associated with low-frequency patterns, we deactivate the neurons identified by the PND and observe the effect on UTSM predictions. For comparison, we also deactivate a set of randomly selected neurons except for the pattern neurons while ensuring that the number of randomly deactivated neurons matches that of the identified pattern neurons at each layer. By measuring the difference in forecasting accuracy, we can

then confirm the importance of pattern neurons. Particularly, if the prediction error increases significantly without pattern neurons, the importance of these neurons to forecasting can be validated:

$$\sum_{d=1}^{D} \|\mathbf{y}_d - \text{UTSM}(\mathbf{x}_d, \boldsymbol{W} \backslash \boldsymbol{w}_{\text{pattern}})\|_1 \gg \sum_{d=1}^{D} \|\mathbf{y}_d - \text{UTSM}(\mathbf{x}_d, \boldsymbol{W} \backslash \boldsymbol{w}_{\text{random}})\|_1 , \quad (5)$$

where $\mathbf{y}_d$ represents the ground truth for the low-frequency sample $\mathbf{x}_d$, and $D$ is the number of verification samples.

**Pattern Neuron Optimizer (PNO).** If pattern neurons are confirmed to exist, the next step is to determine whether optimizing these neurons can enhance urban time series forecasting. To achieve this, we propose a fine-tuning mechanism designed specifically to optimize the detected pattern neurons. The objective of PNO is to minimize this loss while improving forecasting accuracy for low-frequency events, and the loss function is defined as:

$$\mathcal{L}(\hat{\mathbf{y}}, \mathbf{y} \mid \theta_{\boldsymbol{w}_{\text{pattern}}}) = \frac{1}{R} \sum_{r=1}^{R} \|\hat{\mathbf{y}}_r - \mathbf{y}_r\|_1 , \quad (6)$$

where $\theta_{\boldsymbol{w}_{\text{pattern}}}$ represents the parameters associated with the pattern neurons, $\hat{\mathbf{y}}_r$ and $\mathbf{y}_r$ denote the prediction and ground truth for the fine-tuning sample $\mathbf{x}_r$ respectively, and $R$ is the total number of samples used for fine-tuning. The `PN-Train` training algorithm is outlined in Algorithm 1.

## 4 EXPERIMENTS

In this section, we evaluate the capability of our proposed `PN-Train` by designing experiments to address the following questions: **RQ1**: Does `PN-Train` successfully detect the *Pattern Neurons*? **RQ2**: How does `PN-Train` perform in comparison to baseline methods across various urban scenarios by optimizing the detected *Pattern Neurons*? **RQ3**: How does the pattern neuron detector perform compared to existing neuron detection methods? **RQ4**: How do the *Pattern Neurons* in different UTSM components affect forecasting results? **RQ5**: How does `PN-Train` perform under various hyperparameters? **RQ6**: How are the *Pattern Neurons* distributed within the model? **RQ7**: Does `PN-Train` generalize to the broader range of scenarios?

### 4.1 EXPERIMENT SETTINGS

**Datasets**   We perform experiments on two real-world datasets from two urban scenarios: Metro-Traffic (Hogue, 2019) and Pedestrian (Fang et al., 2024). Metro-Traffic contains hourly westbound traffic volumes on Interstate 94 between Minneapolis and St. Paul, MN from 2012 to 2018, including 63 holidays. Pedestrian comprises hourly pedestrian counts from 48 sensors in Melbourne from 2019 to 2022, covering 52 holidays. Detailed dataset statistics are provided in Appendix A.1.

**Baselines**   We evaluate `PN-Train` against nine widely used baselines, categorized as follows: the traditional time series model Historical Average (`HA`); graph-based models including `STGCN` (Yu et al., 2018), `GWNET` (Wu et al., 2019), `AGCRN` (Bai et al., 2020), `PM-MemNet` Lee et al. (2022), and `TESTAM` (Lee & Ko, 2024); and graph-free models including `STID` (Shao et al., 2022), `STWA` (Cirstea et al., 2022), and `STAEformer` (Liu et al., 2023). Detailed baseline descriptions are in Appendix A.2.

**Implementation Details**   All experiments are conducted using PyTorch (Paszke et al., 2019) on a single NVIDIA A100 80GB GPU. The look-back window $L$ and forecasting horizon $H$ are both set to 12. The selective ratio $\epsilon$ is 0.5, with a pattern neuron detection sample length $B$ of 30 and a fine-tuning sample length $R$ of 10. We split the dataset chronologically into training, validation, and test sets in a 6:2:2 ratio. Fine-tuning samples are randomly selected from the holiday data in the training set and are excluded from training to prevent over-training on the same samples. Detection samples are randomly selected from the validation set, while test samples are used for verification. We employ STAEformer (Liu et al., 2023) as our UTSM. During training, the UTSM is optimized using the AdamW optimizer (Loshchilov & Hutter, 2019) with a learning rate $\alpha_1$ of 0.001. Early stopping is applied with a patience of 20 epochs, and the maximum number of epochs is set to 300. For pattern neuron optimization, the UTSM is fine-tuned using the same optimizer with a learning rate $\alpha_2$ of 0.002 for one epoch. Further implementation details can be found in Appendix A.3, while important notations and their parameter settings are in Appendix A.4. The model is evaluated using MAE, RMSE, and WMAPE, with more details in Appendix A.5.

## 4.2 MAIN RESULTS

**Validation of Pattern Neurons**    To address **RQ1** and validate the existence of *Pattern Neurons*, we use PND to detect them and PNV to evaluate `PN-Train`'s performance under neuron deactivation. `Original` leaves all neurons active, `D-PN` deactivates pattern neurons associated with holidays as identified by PND, and `D-Random` deactivates the same number of neurons randomly.

Table 1: Pattern neuron verification via neuron deactivation. Lower MAE, RMSE, and WMAPE values indicate better prediction accuracy. $^†$ denotes statistically worse results.

| Model | Metro-Traffic (Deactivate ratio 7.76%) | | | | | | | | |
|---|---|---|---|---|---|---|---|---|---|
| | Holiday | | | Non-Holiday | | | Overall | | |
| | MAE | RMSE | WMAPE | MAE | RMSE | WMAPE | MAE | RMSE | WMAPE |
| Original | 446.04 | 846.75 | 16.36% | 208.84 | 339.77 | 6.19% | 220.00 | 379.14 | 6.58% |
| D-Random | 492.29 | 833.99 | 18.05% | 263.34 | 380.46 | 7.80% | 274.11 | 413.12 | 8.19% |
| D-PN | 663.46$^†$ | 1046.40$^†$ | 24.33%$^†$ | 474.02$^†$ | 586.01$^†$ | 14.04%$^†$ | 482.93$^†$ | 615.44$^†$ | 14.43%$^†$ |

| Model | Pedestrian (Deactivate ratio 9.77%) | | | | | | | | |
|---|---|---|---|---|---|---|---|---|---|
| | Holiday | | | Non-Holiday | | | Overall | | |
| | MAE | RMSE | WMAPE | MAE | RMSE | WMAPE | MAE | RMSE | WMAPE |
| Original | 109.01 | 259.79 | 29.31% | 78.82 | 196.39 | 21.70% | 80.45 | 200.33 | 22.12% |
| D-Random | 116.99 | 264.03 | 31.46% | 91.75 | 210.79 | 25.26% | 93.12 | 214.01 | 25.60% |
| D-PN | 194.53$^†$ | 370.80$^†$ | 52.31%$^†$ | 174.92$^†$ | 321.45$^†$ | 48.15%$^†$ | 175.98$^†$ | 324.31$^†$ | 48.38%$^†$ |

Results in Table 1 confirm the existence of *Pattern Neurons*, and PND successfully detects them. Deactivating the neurons identified by PND (`D-PN`) leads to a significant performance drop compared to the `Original`, with MAE increasing by 48.75% for the Metro-Traffic dataset and 78.46% for the Pedestrian dataset for holiday samples. In contrast, randomly deactivating an equivalent number of neurons (`D-Random`) causes much smaller degradation: 10.37% for Metro-Traffic and 7.32% for Pedestrian. This stark difference in performance suggests that the neurons detected by PND are indeed closely associated with the patterns of interest, i.e., holidays.

The findings also show that holiday pattern neurons constitute a small fraction of the entire UTSM, comprising 7.76% in the Metro-Traffic dataset and 9.77% in the Pedestrian dataset. Despite their small number, deactivating these pattern neurons significantly degrades performance. Notably, deactivating neurons associated with low-frequency patterns also negatively impacts the performance of non-holiday patterns. This occurs because the pattern neurons include those that capture general time series knowledge critical for all patterns, as they were selected based on their high influence on overall forecasting accuracy. The variation in deactivation ratios between the two datasets demonstrates that our PND can dynamically select neurons based on the data, as it identifies pattern neurons by focusing on those with consistently high attribution scores across all detection samples.

**Overall Performance**    We report the results of `PN-Train` with baselines in Table 2 to answer the **RQ2**. The findings confirm that optimizing the *Pattern Neurons* improves urban time series forecasting. `PN-Train` achieves the best overall performance across both the Metro-Traffic and Pedestrian datasets.

By fine-tuning the holiday pattern neurons, `PN-Train` consistently outperforms `PN-Train` * on both datasets, as it enhances the model's ability to capture holiday patterns. While excluding holiday samples during training causes `PN-Train` * to underperform its base UTSM (`STAEformer`) in the Metro-Traffic dataset, fine-tuning the holiday pattern neurons offsets this and improves forecasting performance. This is because optimizing the holiday neurons helps the network better represent holidays than training on a mix of low-frequency holiday and high-frequency non-holiday samples. In contrast, with more frequent holidays in the Pedestrian dataset, excluding some holiday samples can actually improve accuracy by removing noisy outliers. Nevertheless, fine-tuning the pattern neurons further enhances `PN-Train` *, as holiday events, though more frequent, are still low-frequency overall and may not be fully captured during initial training.

Additionally, fine-tuning the *Pattern Neurons* not only improves performance on holiday samples but also enhances non-holiday and overall performance. This is because these neurons also memorize general time series knowledge, such as level and trend (Brockwell et al., 2016), and optimizing them strengthens the model's representation learning of general time series. This underscores the

Table 2: Comparison with baselines on Metro-Traffic and Pedestrian datasets. Lower MAE, RMSE, and WMAPE indicate better prediction accuracy. * denotes `PN-Train` without PNO. **Best results** are in bold, and second-best are underlined. `PN-Train` employs `STAEformer` as its UTSM.

| | Method | Holiday | | | Non-Holiday | | | Overall | | |
|---|---|---|---|---|---|---|---|---|---|---|
| | | MAE | RMSE | WMAPE | MAE | RMSE | WMAPE | MAE | RMSE | WMAPE |
| Metro-Traffic | HA | 156.64 | 325.58 | 42.12% | 128.28 | 270.90 | 35.31% | 129.82 | 274.14 | 35.69% |
| | STGCN (Yu et al., 2018) | 460.97 | 739.63 | 16.91% | 289.85 | 501.33 | 8.58% | 297.90 | 515.02 | 8.90% |
| | GWNet (Wu et al., 2019) | 534.76 | 832.13 | 19.61% | 347.50 | 582.90 | 10.29% | 356.31 | 596.97 | 10.64% |
| | AGCRN (Bai et al., 2020) | 453.23 | 738.60 | 16.62% | 280.41 | 496.75 | 8.31% | 288.54 | 510.71 | 8.62% |
| | STID (Shao et al., 2022) | 586.90 | 1031.50 | 21.52% | 216.09 | 346.81 | 6.40% | 233.54 | 405.81 | 6.98% |
| | PM-MemNet (Lee et al., 2022) | 554.33 | 916.78 | 20.32% | 375.88 | 666.90 | 11.13% | 384.28 | 680.71 | 11.49% |
| | STWA (Cirstea et al., 2022) | 521.02 | 820.57 | 19.11% | 355.63 | 619.28 | 10.53% | 364.36 | 630.61 | 10.89% |
| | STAEformer (Liu et al., 2023) | 443.23 | 821.42 | 16.25% | 210.41 | 343.01 | 6.23% | 221.37 | 379.29 | 6.62% |
| | TESTAM (Lee & Ko, 2024) | 486.89 | 857.99 | 17.86% | 335.05 | 555.09 | 9.92% | 342.19 | 572.94 | 10.22% |
| | PN-Train * | 446.04 | 846.75 | 16.35% | 208.84 | 339.77 | 6.19% | 220.00 | 379.14 | 6.58% |
| | PN-Train | **430.40** | **816.50** | **15.78%** | **203.62** | **332.15** | **6.03%** | **214.29** | **369.46** | **6.40%** |
| Pedestrian | HA | 208.49 | 388.17 | 64.48% | 255.12 | 471.08 | 83.46% | 253.24 | 468.01 | 82.69% |
| | STGCN (Yu et al., 2018) | 120.75 | 258.53 | 32.47% | 101.61 | 214.32 | 27.97% | 102.65 | 216.95 | 28.22% |
| | GWNet (Wu et al., 2019) | 119.77 | 267.48 | 32.21% | 113.69 | 245.87 | 31.30% | 114.02 | 247.09 | 31.35% |
| | AGCRN (Bai et al., 2020) | 118.48 | 267.32 | 31.86% | 108.22 | 245.55 | 29.79% | 108.78 | 246.78 | 29.91% |
| | STID (Shao et al., 2022) | 116.42 | 263.79 | 31.31% | 85.32 | 206.36 | 23.49% | 87.00 | 209.87 | 23.92% |
| | PM-MemNet (Lee et al., 2022) | 117.44 | 265.09 | 31.58% | 112.18 | 246.64 | 30.88% | 112.48 | 247.69 | 30.92% |
| | STWA (Cirstea et al., 2022) | 114.18 | 261.03 | 30.70% | 106.62 | 234.88 | 29.35% | 106.90 | 236.13 | 29.39% |
| | STAEformer (Liu et al., 2023) | 115.24 | 273.64 | 30.99% | 82.23 | 202.73 | 22.64% | 84.02 | 207.19 | 23.10% |
| | TESTAM (Lee & Ko, 2024) | **103.79** | **257.10** | **27.91%** | 94.04 | 219.46 | 25.89% | 94.57 | 221.67 | 26.00% |
| | PN-Train * | 109.01 | 259.79 | 29.31% | 78.82 | 196.39 | 21.70% | 80.45 | 200.33 | 22.12% |
| | PN-Train | 106.11 | 253.86 | 28.54% | **78.35** | **194.72** | **21.57%** | **79.85** | **198.38** | **21.95%** |

significance of identifying pattern-related neurons to preserve overall performance and even enhance it. Although `TESTAM` performs well on holiday samples in the Pedestrian dataset by leveraging different experts, its overall performance is limited by the routing mechanism. In contrast, `PN-Train` addresses low holiday performance at the neuron level without degrading non-holiday performance, leading to better results across all scenarios.

## 4.3 MODEL ANALYSIS

**Study on Pattern Neuron Detector**  We further assess our proposed PND by comparing it with recent neuron detection techniques to address **RQ3**. Specifically, we evaluate the following variants of `PN-Train`, including: **w/o** PND: excludes the PND; **w** GD: replaces PND with the gradient-based detector from (Chen et al., 2024). **w** FD: replaces PND with the perturbation-based detector from (Zhao et al., 2024). The results are shown in Table 3.

Table 3: Results of `PN-Train` with different neuron detection techniques.

| Model | Metro-Traffic | | | | | | | | |
|---|---|---|---|---|---|---|---|---|---|
| | Holiday | | | Non-Holiday | | | Overall | | |
| | MAE | RMSE | WMAPE | MAE | RMSE | WMAPE | MAE | RMSE | WMAPE |
| **w/o** PND | 1082.46 | 1444.20 | 60.80% | 1294.44 | 1664.49 | 38.34% | 1284.47 | 1654.78 | 38.39% |
| **w** GD | 438.32 | 826.83 | 16.07% | 206.84 | 335.71 | 6.13% | 217.73 | 373.58 | 6.51% |
| **w** FD | 434.76 | 825.71 | 15.94% | 204.52 | 333.76 | 6.05% | 215.36 | 371.80 | 6.44% |
| PN-Train | **430.40** | **816.50** | **15.78%** | **203.62** | **332.15** | **6.03%** | **214.29** | **369.46** | **6.40%** |

| Model | Pedestrian | | | | | | | | |
|---|---|---|---|---|---|---|---|---|---|
| | Holiday | | | Non-Holiday | | | Overall | | |
| | MAE | RMSE | WMAPE | MAE | RMSE | WMAPE | MAE | RMSE | WMAPE |
| **w/o** PND | 238.24 | 426.90 | 64.06% | 225.54 | 378.80 | 62.09% | 226.23 | 381.56 | 62.20% |
| **w** GD | 108.27 | 256.64 | 29.12% | 79.94 | 196.20 | 22.01% | 81.48 | 199.94 | 22.40% |
| **w** FD | 107.55 | 256.26 | 28.92% | 78.93 | 195.39 | 21.73% | 80.48 | 199.16 | 22.13% |
| PN-Train | **106.11** | **253.86** | **28.54%** | **78.35** | **194.72** | **21.57%** | **79.85** | **198.38** | **21.95%** |

The results confirm the importance of neuron detection. Performance drops significantly without it, as fine-tuning all parameters in the UTSM based on low-frequency patterns leads to overfitting. In contrast, UTSMs with neuron detection, i.e., **w** GD, **w** FD, and `PN-Train`, effectively identify and

fine-tune only the pattern neurons, preventing overfitting and preserving the model's generalization capability. It also reveals that perturbation-based detectors outperform gradient-based methods in urban time series forecasting, as they directly measure how changes impact predictions, offering clearer insights into neuron importance. While gradient-based methods capture sensitivity to parameter changes, they fall short in demonstrating a neuron's overall impact on forecasting accuracy. Our proposed PND is a finer-grained perturbation-based detector that evaluates how changes affect predictions at each linear layer in the UTSM, rather than focusing only on attention scores and feed-forward layers as in **w** FD. This allows PND to achieve the best performance.

**Ablation Study** We design the following variants to answer **RQ4** by evaluating the effectiveness of the pattern neuron optimizer (PNO) on different transformer-based UTSM components, including: **w/o** PNO: excludes the PNO in `PN-Train`; **w/o** SD: omits optimization of pattern neurons in the spatial transformer; **w/o** TD: omits optimization of pattern neurons in the temporal transformer; **w/o** AD: omits optimization of pattern neurons in self-attention mechanism; **w/o** FD: omits optimization of pattern neurons in the feed-forward layer.

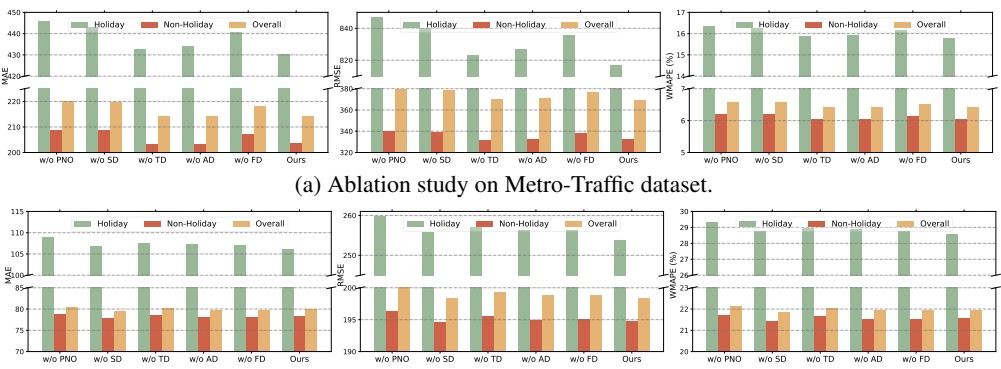

(a) Ablation study on Metro-Traffic dataset.

(b) Ablation study on Pedestrian dataset.

Figure 3: Ablation study results.

The results presented in Figure 3 confirm that PNO significantly enhances forecasting performance. Across both datasets, the absence of PNO leads to a notable decline in accuracy, particularly in holiday scenarios. Fine-tuning the *Pattern Neurons* across all UTSM components proves crucial, as each component addresses a distinct aspect of the data: the spatial transformer learns spatial correlations, the temporal transformer captures temporal patterns, the attention mechanism refines short-term dependencies, and the feed-forward layer enhances long-term memory. `PN-Train` fine-tunes *Pattern Neurons* in all components, consistently outperforming its variants and highlighting the importance of identifying and fine-tuning *Pattern Neurons* across the entire model.

**Hyperparameter Study** We investigate the effects of hyperparameters in `PN-Train` to address **RQ5**. Specifically, we examine three key hyperparameters: the selection ratio ($\epsilon$), the number of detection samples ($B$), and the number of fine-tuning samples ($R$). The results in Figure 4 reveal:

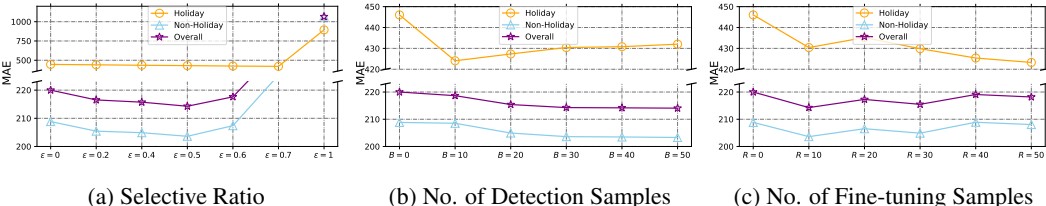

(a) Selective Ratio  (b) No. of Detection Samples  (c) No. of Fine-tuning Samples

Figure 4: Hyperparameter study on Metro-Traffic dataset.

There is a trade-off between holiday and non-holiday performance. When $\epsilon = 1$, all neurons in the UTSM are fine-tuned. Increasing $\epsilon$ from 0 to 0.7 improves holiday performance, but non-holiday performance declines as $\epsilon$ increases from 0.5 to 0.7. This occurs because, with a larger $\epsilon$, too many *Pattern Neurons*, including those responsible for general time series knowledge, are detected and fine-tuned, leading to overfitting the UTSM to holiday patterns. We opt for $\epsilon = 0.5$ as it provides the best balance between holiday and non-holiday performance.

A detection sample size of $B = 30$ is sufficient to identify the *Pattern Neurons*. Increasing the number of detection samples reduces the number of neurons associated with low-frequency patterns being selected, as we only detect neurons with high attribution scores across all detection samples. Consequently, using a larger number of detection samples may cause certain pattern neurons to go undetected due to slight variations in holiday patterns.

`PN-Train` achieves the best performance when $R = 10$, indicating that fine-tuning specific neurons associated with low-frequency patterns is low-cost, requiring only a few samples to boost performance for both low- and high-frequency patterns.

**Pattern Neuron Visualization**    To address **RQ6**, we visualize attribution scores for holiday patterns in the Traffic dataset, which reveals the distribution of *Pattern Neurons* across UTSM layers.

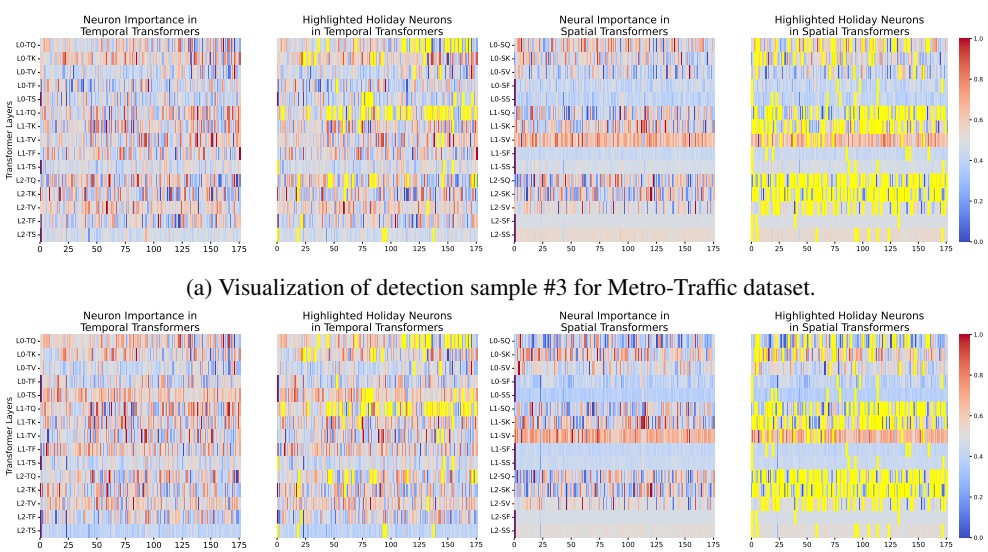

(a) Visualization of detection sample #3 for Metro-Traffic dataset.

(b) Visualization of detection sample #13 for Metro-Traffic dataset.

Figure 5: Visualization of neuron importance, i.e., normalized attribution scores, across USTM layers. LX represents the X-th linear layer in the transformer. TQ, TK, TV, TF, and TS represent neurons in the temporal transformer's query, key, value, first linear layer, and second linear layer, respectively, while SQ, SK, SV, SL, and SF denote the same in the spatial transformer. *Pattern Neurons* are highlighted in yellow.

Figure 5 shows that neurons with high attribution scores consistently appear in similar positions, identifying specific holiday neurons that can be detected with a small number of samples. The number of holiday neurons varies between the temporal and spatial transformers, with more concentrated on the query and key components, emphasizing the role of attention mechanisms in detecting low-frequency patterns like holidays. Furthermore, the distribution of pattern neurons across layers reflects a hierarchical structure, where shallow layers capture general patterns and middle layers refine lower-level features. The visualization of neuron importance for low- and high-frequency patterns, as well as for the Pedestrian dataset, can be found in Appendix A.7.

**Generality Study**    To further evaluate the generality of `PN-Train` for **RQ7**, we present results on the GBAP dataset[1], which spans five years (2017-2021) of traffic flow data and covers both holidays and parades.

The results in Table 4 confirm that `PN-Train` can effectively handle broader urban data beyond traffic and pedestrian counts and can address multiple low-frequency patterns through sequential fine-tuning. It achieves the best overall performance, improving accuracy on holidays and high-frequency patterns by fine-tuning pattern neurons in its UTSM model, i.e., `STAEformer`. While its parade performance is comparable due to `STAEformer`'s limitations, `PN-Train` still achieves an 18.80% MAE improvement, with 18.80% on holidays and 9.86% on parades over baseline mod-

---

[1] GBAP is extracted from the LargeST dataset (Liu et al., 2024)

Table 4: Comparison with baselines on GBAP dataset under holidays and parades.

| Method | Holiday | | | Parade | | | Others | | | Overall | | |
|---|---|---|---|---|---|---|---|---|---|---|---|---|
| | MAE | RMSE | WMAPE | MAE | RMSE | WMAPE | MAE | RMSE | WMAPE | MAE | RMSE | WMAPE |
| HA | 55.63 | 76.11 | 14.88% | 48.09 | 62.53 | 13.03% | 52.13 | 77.60 | 16.88% | 52.18 | 77.55 | 16.83% |
| STGCN | 38.74 | 54.67 | 10.36% | 27.52 | 37.28 | 7.46% | 29.58 | 45.18 | 9.57% | 29.72 | 45.33 | 9.59% |
| GWNet | 42.75 | 61.67 | 11.44% | 27.69 | 38.10 | 7.50% | 31.04 | 47.88 | 10.05% | 31.22 | 48.12 | 10.07% |
| AGCRN | 38.61 | 54.26 | 10.33% | 26.18 | 35.19 | 7.10% | 28.81 | 44.17 | 9.33% | 28.96 | 44.34 | 9.34% |
| STID | 38.25 | 54.29 | 10.23% | 26.28 | 34.84 | 7.12% | 25.43 | 39.85 | 8.23% | 25.64 | 40.11 | 8.27% |
| PM-MemNet | 34.97 | 49.23 | 9.36% | 26.41 | 35.82 | 7.16% | 29.00 | 43.85 | 9.39% | 29.09 | 43.93 | 9.39% |
| STWA | 40.53 | 58.04 | 10.84% | **25.66** | **35.22** | **6.95%** | 30.75 | 46.90 | 9.96% | 30.86 | 47.01 | 9.96% |
| STAEformer | 32.39 | 45.65 | 8.67% | 27.08 | 37.20 | 7.68% | 25.33 | 39.92 | 8.20% | 25.45 | 40.01 | 8.21% |
| TESTAM | 33.86 | 48.04 | 9.06% | 31.98 | 45.34 | 8.66% | 28.08 | 43.52 | 9.09% | 28.18 | 43.60 | 9.09% |
| PN-Train | **32.25** | **45.44** | **8.62%** | 26.73 | 36.68 | 7.24% | **25.25** | **39.81** | **8.18%** | **25.37** | **39.90** | **8.18%** |

els, demonstrating the effectiveness of neuron-level fine-tuning in enhancing UTSM capabilities. Additional results on low-frequency events and datasets are provided in Appendix A.6.

## 5 RELATED WORK

**Urban Time Series Forecasting** is essential for many smart city applications, driving the creation of diverse Urban Time Series Models (UTSMs). Early efforts relied on classical models like ARIMA (Williams & Hoel, 2003; Tran et al., 2015) and Holt-Winters (de Assis et al., 2013; Brügner, 2017), but these methods often struggle to capture the complex patterns inherent in urban data. Recently, deep learning-based models, including graph-based approaches (Zheng et al., 2020; Wu et al., 2019; Bai et al., 2020; Wu et al., 2020), graph-free methods (Deng et al., 2021; Shao et al., 2022; Liu et al., 2023; Wang et al., 2024), and data-adaptive approaches (Chen et al., 2021; Wang et al., 2023; Li et al., 2024; Zhou et al., 2024), have gained prominence for their ability to learn non-linear relationships. However, their focus on overall accuracy often leads to neglect of low-frequency patterns with limited training data (Krawczyk, 2016). Although some works can tackle the imbalanced pattern distribution at the network level using time-varying optimization (Hou et al., 2021), key-value memory retrieval (Lee et al., 2022), or Mixture of Experts frameworks (Lee & Ko, 2024), neuron-level analysis of UTSMs has not been examined. In this work, we investigate neurons associated with low-frequency patterns and confirm that fine-tuning these neurons further enhances the network's forecasting capability.

**Neuron Interpretability** has gained significant attention for explaining neural networks across various applications, from visual (Bau et al., 2017; Mu & Andreas, 2020) to language models (Bau et al., 2019; Xin et al., 2019; Dalvi et al., 2020). Recent studies (Dai et al., 2022; Wang et al., 2022) show that specific neurons in large language models capture knowledge-specific contexts. To detect knowledge neurons, existing methods include gradient-based techniques (Dai et al., 2022; Chen et al., 2024), entropy-based activation analysis (Tang et al., 2024), and perturbation-based difference evaluation (Zhao et al., 2024). While neuron manipulation of detected neurons has improved multilingual capabilities (Tang et al., 2024; Zhao et al., 2024), neuron interpretability in UTSMs remains underexplored. Although ComS2T (Zhou et al., 2024) categorizes neurons as invariant or dynamic, it does not analyze those associated with specific knowledge, making it unsuitable for interpreting pattern-associated neurons. In this work, we employ a fine-grained perturbation-based approach to interpret neurons in UTSMs, revealing the existence of neurons specifically associated with low-frequency patterns in urban time series forecasting.

## 6 CONCLUSION

We introduced PN-Train, a novel training method featuring a perturbation-based neuron detector to confirm pattern neurons in urban time series models. Building on this, we proposed a pattern neuron optimizer that fine-tunes these neurons to improve forecasting for low-frequency patterns, such as holidays. Our experiments showed that fine-tuning less than 10% of the neurons significantly boosts accuracy for these patterns. We also found that in transformer-based urban time series models, the key and query components are critical for capturing patterns. We hope our findings provide a fresh perspective and inspire further exploration of time series models at the neuron level. Future work will study the theoretical foundations of neuron interpretability.

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

## A APPENDIX

### A.1 DATASET DETAILS

Table A.1 provides a summary of the statistical information for the two real-world datasets, Metro-Traffic (Hogue, 2019) and Pedestrian (Fang et al., 2024). This includes the time span of each dataset, the selected frequency and sensor size, as well as the number of weekdays, weekends, and holidays within the time span.

Table A.1: Statistics of the datasets.

| Dataset | Time Span | Frequency | Sensor Size | Weekdays | Weekends | Holidays |
|---|---|---|---|---|---|---|
| Metro-Traffic | 10/02/2012 - 30/09/2018 | 1 hour | 1 | 1,731 | 694 | 63 |
| Pedestrian | 11/02/2019 - 31/10/2022 | 1 hour | 48 | 971 | 388 | 52 |

### A.2 BASELINES

To thoroughly evaluate our model, we compare `PN-Train` with nine widely used urban time series, including the following:

- `HA` is a traditional time series model that forecasts future values by averaging historical data for corresponding time slots.
- `STGCN` (Yu et al., 2018) is a graph-based UTSM that employs graph convolution networks to capture spatial dependencies among citywide sensors and uses a 1D convolution network to model temporal dependencies.
- `GWNET` (Wu et al., 2019) enhances `STGCN` by introducing a self-adaptive graph neural network to learn dynamic spatial dependencies and uses stacked dilated causal convolutions to model temporal patterns.
- `AGCRN` (Bai et al., 2020) is a graph-based model that captures region-specific spatio-temporal correlations through an adaptive graph convolutional recurrent network.
- `STID` (Shao et al., 2022) is a graph-free UTSM that encodes spatial and temporal identities using an embedding layer and applies Multi-Layer Perceptrons to learn spatio-temporal correlations in urban time series data.
- `PM-MemNet` (Lee et al., 2022) uses a key-value memory structure to cluster traffic patterns and dynamically retrieve relevant ones for assisting predictions.
- `STWA` (Cirstea et al., 2022) is a graph-free urban traffic series model (UTSM) that employs location- and time-specific parameters to enable a spatio-temporal aware attention mechanism.
- `STAEformer` (Liu et al., 2023) improves upon `STID` by introducing spatio-temporal adaptive embeddings, allowing the vanilla transformer to learn dynamic spatio-temporal correlations more effectively.
- `TESTAM` (Lee & Ko, 2024) is a graph-based UTSM that captures dynamic spatial relationships through an adaptive graph-based attention mechanism and employs a mixture of experts to capture both regular and irregular patterns in urban time series.

### A.3 EXPERIMENTAL SETUP

All experiments were conducted on an NVIDIA A100 80GB GPU and repeated three times. We used the AdamW optimizer (Loshchilov & Hutter, 2019) with a 0.001 learning rate, early stopping with the patience of 20 epochs and a maximum of 300 epochs. The batch size was 32, with a look-back window ($L$) of 12 and a forecast horizon ($H$) of 12. Implemented in PyTorch (Paszke et al., 2019), our method used the official code for all baselines. `STAEformer` (Liu et al., 2023) served as our UTSM, with all other parameters the same as the original model.

### A.4 NOTATIONS

In this section, we present a table of important notations in Table A.2.

Table A.2: Table of important notations in `PN-Train`.

| Notation | Description | Parameter |
|---|---|---|
| $L$ | Look-back window | 12 |
| $H$ | Forecast horizon | 12 |
| $Attr_p$ | Attribution score for pattern $p$ | - |
| $\epsilon$ | Selective ratio for neurons with high attribution scores | 0.5 |
| $B$ | Sample sizes for pattern neuron detection | 30 |
| $D$ | Sample sizes for pattern neuron verification | - |
| $R$ | Sample sizes for pattern neuron optimization | 10 |

## A.5 METRIC DETAILS

Following the previous studies (Wang et al., 2021; Lee & Ko, 2024), we evaluate performance using three metrics, each assessing the model from a different perspective: Mean Absolute Error (MAE), Root Mean Square Error (RMSE), and Weighted Mean Absolute Percentage Error (WMAPE). MAE measures the average L1 distance between predicted values and the ground truth, making it less sensitive to outliers. RMSE, as the square root of the average L2 distance, gives more weight to outliers. WMAPE evaluates accuracy based on percentage errors, which is scale-independent.

$$\text{MAE} = \frac{1}{\xi}\sum_{i=1}^{\xi}\left|\hat{\mathbf{y}}^i - \mathbf{y}^i\right|, \text{RMSE} = \sqrt{\frac{1}{\xi}\sum_{i=1}^{\xi}\left(\hat{\mathbf{y}}^i - \mathbf{y}^i\right)^2}, \text{WMAPE} = \frac{\sum_{i=1}^{\xi}\left|\hat{\mathbf{y}}^i - \mathbf{y}^i\right|}{\sum_{i=1}^{\xi}|\mathbf{y}^i|} \quad (7)$$

where $\hat{\mathbf{y}}^i$ and $\mathbf{y}^i$ denote the predicted values and ground truth, and $\xi$ is the total number of samples.

To ensure a comprehensive evaluation across all patterns, we evaluate the forecasting performance separately for low-frequency events, high-frequency events, and overall.

## A.6 FORECASTING PERFORMANCE UNDER VARIOUS LOW-FREQUENCY EVENTS

We have discussed `PN-Train` for well-defined low-frequency events, such as holidays, which are critical for smart city applications (Cools et al., 2007; McElroy et al., 2018; Luo et al., 2019b), in Section 4.2 and explore its generalization in Section 4.3. This section demonstrates that `PN-Train` is also capable of handling various low-frequency events, including unpredictable ones such as extreme weather[2].

Table A.3: Comparison with baselines on Metro-Traffic and Pedestrian datasets under holidays and extreme weather conditions. `PN-Train` employs `STAEformer` as its UTSM.

| | Method | Holiday | | | Extreme Weather | | | Others | | | Overall | | |
|---|---|---|---|---|---|---|---|---|---|---|---|---|---|
| | | MAE | RMSE | WMAPE | MAE | RMSE | WMAPE | MAE | RMSE | WMAPE | MAE | RMSE | WMAPE |
| **Metro-Traffic** | HA | 977.20 | 1347.43 | 38.64% | 1471.16 | 1844.52 | 73.38% | 736.87 | 1133.51 | 21.87% | 744.57 | 1141.54 | 22.25% |
| | STGCN | 502.70 | 777.93 | 19.88% | 1261.39 | 1688.29 | 62.92% | 290.26 | 499.96 | 8.61% | 297.90 | 515.02 | 8.90% |
| | GWNet | 577.56 | 885.53 | 22.84% | 1407.70 | 1837.06 | 70.21% | 348.03 | 581.17 | 10.33% | 356.31 | 596.97 | 10.65% |
| | AGCRN | 485.42 | 762.78 | 19.20% | 1279.55 | 1760.73 | 63.82% | 281.03 | 495.32 | 8.34% | 288.54 | 510.71 | 8.62% |
| | STID | 596.58 | 1027.33 | 23.59% | 1044.86 | 1506.94 | 52.12% | 222.37 | 370.10 | 6.60% | 233.54 | 405.81 | 6.98% |
| | PM-MemNet | 613.67 | 975.18 | 24.27% | 1204.70 | 1663.15 | 60.09% | 376.41 | 667.21 | 11.17% | 384.28 | 680.71 | 11.49% |
| | STWA | 518.14 | 783.14 | 20.49% | 1340.30 | 1778.30 | 66.85% | 357.02 | 619.36 | 10.60% | 363.72 | 630.53 | 10.87% |
| | STAEformer | 409.43 | 721.50 | 16.19% | 1011.52 | 1511.97 | 50.45% | 214.60 | 358.94 | 6.37% | 221.37 | 379.29 | 6.62% |
| | TESTAM | 453.82 | 734.22 | 17.94% | 1256.48 | 1791.29 | 62.67% | 337.00 | 561.62 | 10.00% | 342.19 | 572.94 | 10.22% |
| | PN-Train | **406.33** | **718.56** | **16.00%** | **966.82** | **1454.16** | **48.22%** | **213.15** | **357.59** | **6.33%** | **219.77** | **377.33** | **6.57%** |
| **Pedestrian** | HA | 159.64 | 335.08 | 41.87% | 109.25 | 220.41 | 34.38% | 129.01 | 272.33 | 35.46% | 129.82 | 274.14 | 35.69% |
| | STGCN | 129.01 | 280.76 | 33.84% | 109.46 | 208.92 | 34.45% | 101.55 | 214.29 | 27.91% | 102.65 | 216.95 | 28.22% |
| | GWNet | 128.51 | 293.30 | 33.71% | 129.28 | 259.03 | 40.69% | 113.23 | 244.98 | 31.13% | 114.02 | 247.09 | 31.35% |
| | AGCRN | 122.53 | 288.24 | 32.14% | 105.31 | 202.70 | 33.14% | 108.31 | 245.70 | 29.77% | 108.78 | 246.78 | 29.91% |
| | STID | 123.65 | 290.17 | 32.43% | **85.96** | **171.74** | **27.05%** | 85.63 | 206.77 | 23.54% | 87.00 | 209.87 | 23.92% |
| | PM-MemNet | 122.37 | 286.36 | 32.10% | 107.86 | 210.53 | 33.95% | 112.18 | 246.64 | 30.84% | 112.48 | 247.69 | 30.92% |
| | STWA | 120.30 | 286.14 | 31.55% | 103.03 | 197.56 | 32.43% | 106.46 | 234.61 | 29.26% | 106.79 | 236.09 | 29.36% |
| | STAEformer | 120.15 | 296.71 | 31.51% | 97.76 | 193.38 | 30.77% | 82.44 | 203.25 | 22.66% | 84.02 | 207.19 | 23.10% |
| | TESTAM | **111.29** | 286.88 | **29.19%** | 116.99 | 235.39 | 36.82% | 93.59 | 218.60 | 25.73% | 94.57 | 221.67 | 26.00% |
| | PN-Train | 115.85 | **283.23** | 30.39% | 89.89 | 178.33 | 28.29% | **79.80** | **196.79** | **21.93%** | **81.24** | **200.28** | **22.34%** |

---

[2]https://open-meteo.com/en/docs/historical-weather-api

In Table A.3, we present the forecasting results of `PN-Train` during two types of low-frequency events, including holidays and extreme weather. The results demonstrate that `PN-Train` effectively tackles multiple low-frequency patterns and significantly outperforms baseline methods in overall performance. It enhances overall forecasting accuracy, achieving 36.95% MAE improvement on the Metro-Traffic dataset and 22.23% on the Pedestrian dataset. Specifically, it improves MAE by 28.78% for holidays and 23.08% for extreme weather on the Metro-Traffic dataset, while achieving 8.33% improvement for holidays and 3.58% for extreme weather on the Pedestrian dataset. This success stems from its perturbation-based detector, which identifies neurons associated with specific patterns, enabling targeted fine-tuning. By leveraging relevant samples, `PN-Train` achieves robust performance in managing low-frequency patterns. Unlike existing methods (Lee et al., 2022; Lee & Ko, 2024) that rely on external features (e.g., time of day, day of the week, time step), `PN-Train` fine-tunes neurons using historical patterns, reducing reliance on external features. This approach enhances generalization and ensures superior performance in complex scenarios.

## A.7 PATTERN NEURON VISUALIZATION

We further visualize pattern neurons for both low- and high-frequency patterns. Specifically, we visualize holiday, non-holiday, and holiday-specific neurons on the Metro-Traffic dataset in Figure A.1.

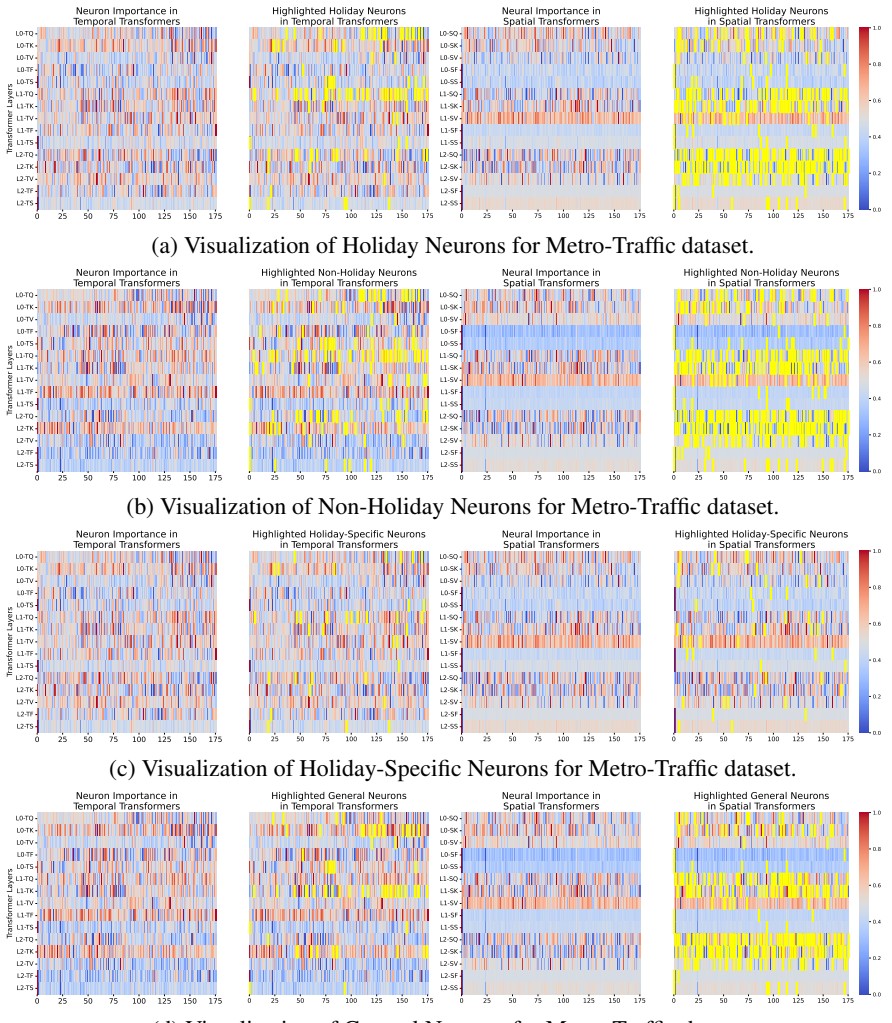

(a) Visualization of Holiday Neurons for Metro-Traffic dataset.

(b) Visualization of Non-Holiday Neurons for Metro-Traffic dataset.

(c) Visualization of Holiday-Specific Neurons for Metro-Traffic dataset.

(d) Visualization of General Neurons for Metro-Traffic dataset.

Figure A.1: Visualization of neuron importance, i.e., normalized attribution scores, across USTM layers for the Metro-Traffic dataset. *Pattern Neurons* are highlighted in yellow.

The results confirm our assumption that holiday neurons include those specific to low-frequency events, which do not contribute to high-frequency events, as well as those that learn general time series features useful for both low- and high-frequency patterns. Additionally, pattern neurons are primarily located in the transformer's query and key components, which are responsible for capturing patterns (Geshkovski et al., 2023).

Furthermore, in Figure A.2, we visualize pattern neurons for the holiday pattern on the Pedestrian dataset. Similar to the holiday neurons in the Metro-Traffic dataset discussed in the main paper, we observe that high attribution scores consistently appear in similar positions, with the query and key components playing a crucial role in emphasizing holiday patterns.

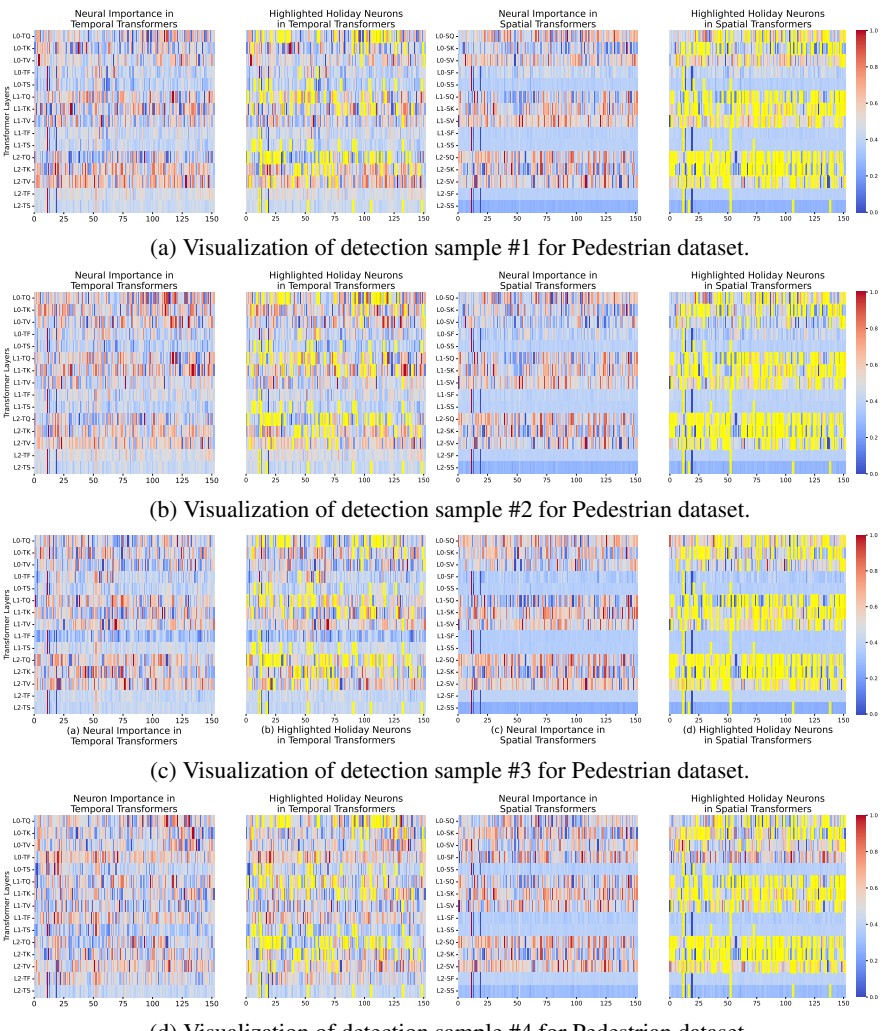

(a) Visualization of detection sample #1 for Pedestrian dataset.

(b) Visualization of detection sample #2 for Pedestrian dataset.

(c) Visualization of detection sample #3 for Pedestrian dataset.

(d) Visualization of detection sample #4 for Pedestrian dataset.

Figure A.2: Visualization of neuron importance, i.e., normalized attribution scores, across USTM layers for the Pedestrian dataset. *Pattern Neurons* are highlighted in yellow.

