# OpenReview forum: "Investigating Pattern Neurons in Urban Time Series Forecasting"
_ICLR.cc/2025/Conference — ICLR 2025 Poster_

### Official Review · Reviewer_T7Mg · 2024-10-21

**Soundness:** 3
**Presentation:** 4
**Contribution:** 3
**Rating:** 6
**Confidence:** 5

**Summary:**

This work addresses the challenge of predicting low-frequency events (e.g., holidays, emergencies) in urban time series data, which are often overlooked by traditional models. The authors propose Pattern Neuron guided Training (PN-Train), a novel method that identifies and fine-tunes neurons specifically associated with low-frequency patterns. PN-Train incorporates a perturbation-based Pattern Neuron Detector (PND) to locate key neurons, and a Pattern Neuron Optimizer (PNO) to enhance their learning without compromising high-frequency event predictions.

**Strengths:**

S1: PN-Train introduces a novel approach by using a perturbation-based detector to identify neurons associated with low-frequency patterns and fine-tuning these neurons to improve prediction accuracy. This is an innovative perspective, combining neuron-level analysis with practical forecasting tasks.

S2: By capturing and fine-tuning neurons under different patterns, PN-Train enhances the fairness of predictions for low-frequency events, reduces model bias, and helps in forecasting scenarios with sparse data.

S3: The experiments provide solid evidence of the existence of Pattern Neurons, with case studies offering visualized results that make the concept highly intuitive and easy to understand.

**Weaknesses:**

W1: While the paper introduces neuron detection and optimization methods, it lacks in-depth theoretical explanation for why these neurons exhibit distinct behavior for low-frequency events. More analysis and discussion[1,2] on the working mechanisms of pattern neurons, such as potential differences in weight distribution or activation between high- and low-frequency events, is needed, along with supporting theoretical calculations.

W2: The paper highlights patterns for holidays and emergencies in the dataset, but in real-world applications, low-frequency events can be more random and harder to predict. Further discussion on how PN-Train handles unpredictable low-frequency events, such as concert events, sport events, or extreme weather like typhoon,  and then testing its generalization ability in more complex scenarios would be beneficial.

W3: There are some typos, such as inconsistent citation formatting in reference 766, the use of "$B" in Figure 5b, and inconsistency between "B" in Figure 5 and "D" in line 486. Please clearfully check for typos throughout the paper.


[1] Zhou Z, Huang Q, Wang B, et al. ComS2T: A complementary spatiotemporal learning system for data-adaptive model evolution[J]. arXiv preprint arXiv:2403.01738, 2024.

[2] Tang T, Luo W, Huang H, et al. Language-specific neurons: The key to multilingual capabilities in large language models. ACL, 2024

**Questions:**

Please see W1-W3.

---

> ### Author Response · Authors · 2024-11-22
>
> > **W1: While the paper introduces neuron detection and optimization methods, it lacks in-depth theoretical explanation for why these neurons exhibit distinct behavior for low-frequency events. More analysis and discussion [1,2] on the working mechanisms of pattern neurons, such as potential differences in weight distribution or activation between high- and low-frequency events, is needed, along with supporting theoretical calculations.**
>
> **[Mechanisms]** ComS2T [1], LAPE [2], and PN-Train differ in both methodology and application. ComS2T [1], from a complementary learning systems perspective, disentangles neurons into stable (invariant) and dynamic (adaptive) categories using variation matrices, enabling adaptability to dynamic spatiotemporal data without specifically addressing low-frequency patterns. In contrast, LAPE [2] and PN-Train take a pattern-based perspective, identifying neurons linked to specific patterns. LAPE uses activation probabilities to detect pattern-specific neurons for large language model understanding but does not fine-tune these neurons for improved performance. PN-Train, however, identifies pattern-related neurons using activation values and fine-tunes them to improve performance on low-frequency patterns, while also preserving or even enhancing accuracy on high-frequency patterns.
>
>
> **[Weight Distribution]** We have added visualizations of pattern neurons in Figure A.1 (also available at https://anonymous.4open.science/r/PN-Train/figures/vis.pdf), showing the distribution of holiday, non-holiday, and holiday-specific neurons. These visualizations confirm our assumption that holiday neurons include those specific to low-frequency events, which do not contribute to high-frequency events, and those that learn general time series features useful for both low- and high-frequency patterns. Additionally, pattern neurons are primarily located in the transformer's query and key components for pattern capturing.
>
> **[Theoretical Explanation]** Understanding neurons in deep models remains challenging, with existing strategies [2, 3, 4, 5] relying on empirical validation. Similarly, PN-Train uses validation and visualization to confirm that identified neurons capture distinct behaviors for specific patterns. Empirical results show that deactivating these neurons significantly impairs low-frequency pattern forecasting. Notably, these neurons are primarily located in the transformer's query and key components, which are responsible for pattern clustering [6].
>
> Furthermore, we provide a preliminary explanation for why these neurons exhibit distinct behaviors in response to low-frequency events. The optimization process in deep networks allows neurons to specialize by aligning their weights with specific data features.
> For a neuron $h^k_i = f(x, w^k_i)$, optimization minimizes the loss through backpropagation, where the weight update follows $\Delta w^k_i \propto -\frac{\partial L}{\partial w^k_i}$. When a specific input pattern $x^p$ significantly impacts the loss, $w^k_i$ aligns with $x^p$ to minimize the error. This alignment enables the neuron $h^k_i$ to effectively capture the pattern $p$ after training with $x^p$. Building on this behavior, our approach identifies neurons by quantifying the influence of the $k$-th neuron on a specific low-frequency pattern $p$, thereby directly pinpointing neurons associated with $p$.
>
> While we acknowledge the importance of theoretical calculations, they often rely on strong assumptions. Given that our primary contribution focuses on identifying and fine-tuning pattern neurons to enhance urban time series forecasting, we prioritize empirical validation in this work, leaving deeper theoretical exploration for future studies.

---

> > ### Author Response · Authors · 2024-11-22
> >
> > > **W2: The paper highlights patterns for holidays and emergencies in the dataset, but in real-world applications, low-frequency events can be more random and harder to predict. Further discussion on how PN-Train handles unpredictable low-frequency events, such as concert events, sport events, or extreme weather like typhoons, and then testing its generalization ability in more complex scenarios would be beneficial.**
> >
> > PN-Train can handle unpredictable low-frequency events by fine-tuning the network with historically relevant data. Its perturbation-based detector identifies neurons associated with specific patterns, enabling fine-tuning tailored to the target pattern. With relevant samples, the network can be fine-tuned to achieve strong performance in handling unpredictable patterns.
> >
> > Below, we present the forecasting results of  PN-Train during extreme weather [7]. The results show that PN-Train significantly outperforms state-of-the-art baselines under extreme weather conditions. Unlike existing methods that rely on external features like time of day, day of the week, and time step, PN-Train handles unpredictable low-frequency events by fine-tuning neurons with historical patterns. This approach reduces dependence on external features, improves generalization, and delivers superior performance in complex, unpredictable scenarios.
> >
> >
> > **Metro-Traffic**:
> >
> > |Method||Holiday|||Extreme Weather|||Others|||Overall||
> > |:-:|:-:|:-:|:-:|:-:|:--:|:-:|:-:|:-:|:-:|:-:|:-:|:-:|
> > ||MAE|RMSE|WMAPE|&nbsp;MAE|RMSE|WMAPE&nbsp;|&nbsp;MAE|RMSE|WMAPE|&nbsp;MAE|RMSE|WMAPE|
> > |STAEformer|409.43|721.50|16.19%&nbsp;|&nbsp;1011.52|1511.97|50.45%&nbsp;|&nbsp;214.60|358.94|6.37%&nbsp;|&nbsp;221.37|379.29|6.61%|
> > |TESTAM|453.82|734.22|17.94%&nbsp;|&nbsp;1256.48|1791.29|62.67%&nbsp;|&nbsp;337.00|561.62|10.00%&nbsp;|&nbsp;342.19|572.94|10.22%|
> > |PN-Train|**406.33**|**718.56**|**16.00%**&nbsp;|&nbsp;**966.82**|**1454.16**|**48.22%**&nbsp;|&nbsp;**213.15**|**357.59**|**6.33%**&nbsp;|&nbsp;**219.77**|**377.33**|**6.57%**|
> >
> >
> > **Pedestrian**:
> >
> > |Method||Holiday|||Extreme Weather|||Others|||Overall||
> > |:-:|:-:|:-:|:-:|:-:|:--:|:-:|:-:|:-:|:-:|:-:|:-:|:-:|
> > ||MAE|RMSE|WMAPE&nbsp;|&nbsp;MAE|RMSE|WMAPE&nbsp;|&nbsp;MAE|RMSE|WMAPE&nbsp;|&nbsp;MAE|RMSE|WMAPE|
> > |STAEformer|120.15|296.71|31.51%&nbsp;|&nbsp;97.76|193.38|30.77%&nbsp;|&nbsp;82.44|203.25|22.66%&nbsp;|&nbsp;84.02|207.19|23.10%|
> > |TESTAM|**111.29**|286.88|**29.19%**&nbsp;|&nbsp;116.99|235.39|36.82%&nbsp;|&nbsp;93.59|218.60|25.73%&nbsp;|&nbsp;94.57|221.67|26.00%|
> > |PN-Train|115.85|**283.23**|30.39%&nbsp;|&nbsp;**89.89**|**178.33**|**28.29%**&nbsp;|&nbsp;**79.80**|**196.79**|**21.93%**&nbsp;|&nbsp;**81.24**|**200.28**|**22.34%**|
> >
> >
> > > **W3: Typos**
> >
> > Thank you for pointing them out. We have corrected all typos and carefully reviewed the paper for any inconsistencies in the revised version.
> >
> > ---
> >
> > We sincerely thank you for your constructive comments. Your insights have been invaluable in improving our manuscript. We hope our responses have addressed your concerns. If any queries remain, we would be happy to engage in further discussions and will update the revision accordingly as soon as possible.
> >
> > ---
> >
> >
> > Reference:
> >
> > [1] Zhou Z, Huang Q, Wang B, et al. ComS2T: A complementary spatiotemporal learning system for data-adaptive model evolution[J]. arXiv preprint arXiv:2403.01738, 2024.
> >
> > [2] Tang T, Luo W, Huang H, et al. Language-specific neurons: The key to multilingual capabilities in large language models. ACL, 2024
> >
> > [3] Dai, Damai, et al. "Knowledge neurons in pretrained transformers." ACL (2022).
> >
> > [4] Voita, Elena, Javier Ferrando, and Christoforos Nalmpantis. "Neurons in large language models: Dead, n-gram, positional."  ACL (Findings) 2024.
> >
> > [5] Chen, Yuheng, et al. "Journey to the center of the knowledge neurons: Discoveries of language-independent knowledge neurons and degenerate knowledge neurons." AAAI 2024.
> >
> > [6] Geshkovski, Borjan, et al. "A mathematical perspective on transformers." arXiv preprint arXiv:2312.10794 (2023).
> >
> > [7] https://open-meteo.com/en/docs/historical-weather-api

---

> > > ### Comment · Reviewer_T7Mg · 2024-11-25
> > > **Thanks for rebuttal**
> > >
> > > Thank you for rebuttal. Some of my concerns have been addressed. Actually, I think this work shares similarity with continuous learning as well as neuro disentanglement and fine-tuning learning. More discussions should be included in the manuscript and the significance of emphasizing the low-frequency events should be appropriately explained.
> > > Thus, I will keep my score with above threshold.

---

> > > > ### Author Response · Authors · 2024-11-25
> > > >
> > > > Thank you very much for supporting our paper and providing insightful comments to enhance its quality. We will diligently incorporate your suggestions into the revised version, including a detailed discussion on the differences with other methods and the significance of emphasizing the low-frequency events. We sincerely appreciate your time and effort.

---

> > > > > ### Author Response · Authors · 2024-11-26
> > > > >
> > > > > Thank you once again for your insightful comments on our manuscript. Your valuable suggestions are truly inspiring and have greatly enhanced the quality of our work. We have carefully incorporated your suggestions into the revised version, particularly in Sections 1 and 5, with the changes highlighted in red. Below, we provide more discussions:
> > > > >
> > > > > **[Detailed discussion on the differences with with other methods]** Continuous learning [1, 2], fine-tuning learning [3, 4], and PN-Train all fine-tune pretrained networks to enhance forecasting. However, the former methods focus on adapting the network for new data and optimizing all patterns simultaneously. While these methods improve forecasting, they often overlook low-frequency patterns due to limited relevant data. In contrast, PN-Train directly fine-tunes pattern neurons associated with low-frequency patterns, reducing forecasting errors for such events while maintaining strong overall performance.
> > > > >
> > > > > Neuron disentanglement [4] and PN-Train both categorize neurons into distinct groups; however, they differ fundamentally. The former employs additional architectural components to disentangle neurons into invariant and adaptive categories for data adaptation. In contrast, PN-Train leverages relevant samples to directly identify and fine-tune pattern neurons, enabling a better understanding of neurons associated with specific patterns.
> > > > >
> > > > > **[Significance of emphasizing the low-frequency events]** Accurate forecasting for low-frequency events, such as holidays, is essential because these events significantly deviate from regular patterns, impacting traffic management and resource allocation in urban systems. For instance, travel demand predictions for holidays like Thanksgiving have consistently drawn attention from the Department of Transportation [5], car rental companies [6], and ride-hailing platforms [7]. Precise predictions in such events ensure optimized operations, reduced costs, and improved user experiences.
> > > > >
> > > > > ---
> > > > > We hope our revised version addresses your concerns. We sincerely appreciate your time and effort.
> > > > >
> > > > > ---
> > > > >
> > > > > Reference:
> > > > >
> > > > >
> > > > >
> > > > >
> > > > > [1] Chen, Xu, Junshan Wang, and Kunqing Xie. Trafficstream: A streaming traffic flow forecasting framework based on graph neural networks and continual learning. IJCAI, 2021.
> > > > >
> > > > >
> > > > > [2] Wang, Binwu, et al. Pattern expansion and consolidation on evolving graphs for continual traffic prediction. KDD, 2023.
> > > > >
> > > > >
> > > > > [3] Li, Zhonghang, et al. FlashST: A Simple and Universal Prompt-Tuning Framework for Traffic Prediction. ICML, 2024.
> > > > >
> > > > >
> > > > > [4] Zhou Z, Huang Q, Wang B, et al. ComS2T: A complementary spatiotemporal learning system for data-adaptive model evolution[J]. arXiv preprint arXiv:2403.01738, 2024.
> > > > >
> > > > >
> > > > > [5] U.S. Department OF Transportation. Fact sheet: Biden-harris administration efforts to ensure air travel is safe, efficient, and fair, 2024. URL https://www.transportation.gov/briefing-room/fact-sheet-biden-harris-administration-efforts-ensure-air-travel-safe-efficient-and
> > > > >
> > > > >
> > > > > [6] American Automobile Association. Nearly 80 million americans expected to travel for thanksgiving, 2024. URL https://minneapolis.aaa.com/news/nearly-80-million-americans-expected-travel-thanksgiving.
> > > > >
> > > > >
> > > > > [7] John Mark Nickels. New features to make holiday travel more effortless, 2024. URL https://www.uber.com/newsroom/airport-travel/.

---

### Official Review · Reviewer_3uYm · 2024-11-03

**Soundness:** 4
**Presentation:** 3
**Contribution:** 3
**Rating:** 6
**Confidence:** 2

**Summary:**

This paper firstly investigates how urban time series models (UTSMs) can be utilized to deal with the low-frequence events in urban management. It proposes a pattern neuron-guided training (PN-Train) method consisting of a perturbation-based detector and a fine-tuning mechanism. The experiments on a series of time series forecasting tasks show the superiority of PN-Train, which enhances performance in low-frequency events while maintaining high performance in high-frequency events.

**Strengths:**

[1] A simple yet effective method is proposed to deal with the low-frequency patterns in urban time series data. It is of certain novelty to design a method from the neural aspect, making

[2] Experiments on multiple datasets validate the existence of pattern neurons and the proposed PND stage successfully detects them. Overall, the empirical studies are solid.

[3] The paper is well-written and easy to follow, with sufficient technical details on the pattern neuron detector, pattern neuron verifier, and pattern neuron optimizer.

**Weaknesses:**

[1] It is better to have a more complete discussion about low-frequency events, e.g., how these events are handled in previous works (Related Works section), how low-frequency events should be technically defined, and how the evaluation metrics are established to verify the effectiveness of the methods.

[2] Lack of comparison with baseline methods that deal with various unbalanced data distribution problems.

**Questions:**

Q1. Is there any way to quantitatively define or model the frequency of events?

Q2. Is there any theoretical principle that can support the neuron-based strategy?

---

> ### Author Response · Authors · 2024-11-22
>
> We sincerely appreciate your insightful review and comments. Below, we provide detailed responses to the specific concerns you raised.
>
> > **Q1: Is there any way to quantitatively define or model the frequency of events?**
>
> A low-frequency event can be defined as a specific type of event that occurs infrequently, with an occurrence rate below a predefined threshold. We quantify this using the frequency rate of the event [1], calculated as $(\text{number of event days} / \text{total observation days}) \times$ 100%.
>
> In this study, holidays are considered low-frequency events due to their limited annual occurrence, accounting for approximately 3.01% (11 days out of 365 or 366) in the metro-traffic dataset and 3.56% (13 days out of 365 or 366) in the pedestrian dataset.
>
> > **W1: It is better to have a more complete discussion about low-frequency events, e.g., how these events are handled in previous works (Related Works section), how low-frequency events should be technically defined, and how the evaluation metrics are established to verify the effectiveness of the methods.**
>
> **[Low-frequency events]** While existing studies recognize the importance of accurate forecasting for holidays [2, 3, 4, 5, 6], they primarily focus on improving accuracy for all patterns simultaneously, without explicitly distinguishing between low-frequency and high-frequency events. Among these, [3] classifies holidays and non-holidays as special periods and normal periods, respectively. We classify holidays as low-frequency events because they are infrequent, comprising less than 4% of the year [1].
>
> **[Related Works]** Previous works address low-frequency events, such as holidays, at the network level. Early approaches [2, 3] use feature extraction to encode factors like holidays, but they overlook the imbalance between low- and high-frequency events, leading to degraded prediction performance for low-frequency events. Recent methods introduce adaptive mechanisms to address this. For example, STV [4] embeds holiday names and corrects bias toward normal-day patterns by distinguishing their distributions through a two-stage training process. PM-MemNet [5] uses a key-value memory structure to cluster traffic patterns and dynamically retrieve relevant ones for accurate predictions. TESTAM [6] leverages a Mixture of Experts to select the most suitable expert for low-frequency events, improving predictions through specialized routing. However, these methods rely on additional network components to handle low-frequency patterns. In contrast, our work improves the network at the component level by directly fine-tuning pattern neurons, without the need for extra network structures.
>
> **[Evaluation Metrics]**: We used well-established metrics [2, 3, 6], including MAE, RMSE, and WMAPE, to verify the effectiveness of the methods. We evaluate performance separately for holidays (low-frequency events), non-holidays (high-frequency events), and overall, ensuring a comprehensive evaluation across all scenarios.
>
> > **Q2: Is there any theoretical principle that can support the neuron-based strategy?**
>
> Understanding neurons in deep models remains challenging, and existing pattern-based strategies [7, 8, 9, 10] primarily rely on empirical validation. Similarly, PN-Train utilizes empirical methods to support its neuron-based strategy through the detection and visualization of pattern neurons. Empirical results confirm that these neurons are successfully detected and are primarily located in the query and key components. This aligns with the theoretical principle on transformers, which suggests that the query and key components are responsible for pattern clustering [11].
>
> Additionally, we provide a preliminary theoretical explanation from an optimization perspective to support such neuron-based strategies. In deep networks, optimization allows neurons to specialize by aligning their weights with specific data features. For a neuron $h^k_i = f(x, w^k_i)$, weights are updated via backpropagation following $\Delta w^k_i \propto -\frac{\partial L}{\partial w^k_i}$. When a specific input pattern $x^p$ significantly affects the loss, $w^k_i$ aligns with $x^p$ to minimize the error. This alignment allows the neuron $h^k_i$ to effectively capture the pattern $p$ after training with $x^p$. Based on this behavior, neuron-based strategies quantify the influence of the $k$-th neuron on a given pattern $p$, enabling the direct identification of neurons associated with pattern $p$.
>
> While we acknowledge the importance of theoretical principles, they often rely on strong assumptions. Given that our primary contribution focuses on identifying and fine-tuning pattern neurons to enhance urban time series forecasting, we prioritize empirical validation in this work, leaving deeper theoretical exploration for future studies.

---

> ### Author Response · Authors · 2024-11-22
>
> > **W2: Lack of comparison with baseline methods that deal with various unbalanced data distribution problems.**
>
> We have included an additional baseline, PM-MemNet [5], which can tackle unbalanced data distributions by clustering time series patterns into 100 groups within a memory bank to assist the predictions. We have also compared our model with TESTAM [6], which uses a Mixture of Experts to dynamically select the most suitable expert for handling diverse unbalanced distributions based on input patterns.
>
> The results below show that PN-Train outperforms existing methods in handling unbalanced data distributions. PM-MemNet performs worse than TESTAM and PN-Train because it relies heavily on fixed memory patterns extracted from the training set, which may include noise, reducing its forecasting accuracy. TESTAM performs well on low-frequency patterns like holidays in the Pedestrian dataset thanks to its routing mechanism. However, its overall performance is limited because unnecessary routing for dominant patterns reduces its effectiveness on high-frequency events.
>
> **Metro-Traffic**:
>
> |Method||Holiday|||Non-Holiday|||Overall||
> |:-:|:-:|:-:|:-:|:-:|:-:|:-:|:-:|:-:|:-:|
> ||MAE|RMSE|WMAPE&nbsp;&nbsp;|&nbsp;&nbsp;MAE|RMSE|WMAPE&nbsp;&nbsp;|&nbsp;&nbsp;MAE|RMSE|WMAPE|&nbsp;&nbsp;MAE|RMSE|WMAPE|
> |PM-MemNet|554.33|916.78|20.32%&nbsp;&nbsp;|&nbsp;&nbsp;375.88|666.90|11.13%&nbsp;&nbsp;|&nbsp;&nbsp;384.28|680.71|11.49%&nbsp;&nbsp;|&nbsp;&nbsp;384.28|680.71|11.49%|
> |TESTAM|486.89|857.99|17.86%&nbsp;&nbsp;|&nbsp;&nbsp;335.05|555.09|9.92%&nbsp;&nbsp;|&nbsp;&nbsp;342.19|572.94|10.22%&nbsp;&nbsp;|&nbsp;&nbsp;342.19|572.94|10.22%|
> |PN-Train|**430.40**|**816.50**|**15.78%**&nbsp;&nbsp;|&nbsp;&nbsp;**203.62**|**332.15**|**6.03%**&nbsp;&nbsp;|&nbsp;&nbsp;**214.29**|**369.46**|**6.40%**&nbsp;&nbsp;|&nbsp;&nbsp;**214.29**|**369.46**|**6.40%**|
>
> **Pedestrian**:
>
> |Method||Holiday|||Non-Holiday|||Overall||
> |:-:|:-:|:-:|:-:|:-:|:-:|:-:|:-:|:-:|:-:|
> ||MAE|RMSE|WMAPE&nbsp;&nbsp;|&nbsp;&nbsp;MAE|RMSE|WMAPE&nbsp;&nbsp;|&nbsp;&nbsp;MAE|RMSE|WMAPE|
> |PM-MemNet|117.44|265.09|31.58%&nbsp;&nbsp;|&nbsp;&nbsp;112.18|246.64|30.88%&nbsp;&nbsp;|&nbsp;&nbsp;112.48|247.69|30.92%|
> |TESTAM|**103.79**|**257.10**|**27.91%**&nbsp;&nbsp;|&nbsp;&nbsp;94.04|219.46|25.89%&nbsp;&nbsp;|&nbsp;&nbsp;94.57|221.67|26.00%|
> |PN-Train|106.11|253.86|28.54%&nbsp;&nbsp;|&nbsp;&nbsp;**78.35**|**194.72**|**21.57%**&nbsp;&nbsp;|&nbsp;&nbsp;**79.85**|**198.38**|**21.95%**|
>
> ---
>
> We sincerely thank you for your constructive comments. Your insights have been invaluable in improving our manuscript. We hope our responses have addressed your concerns and have been incorporated into the revised version, with changes marked in **RED**. If any questions remain, we would be happy to engage in further discussions.
>
> ---
>
>
> **Reference**:
>
> [1] Devore, Jay L. "Probability and statistics." Pacific Grove: Brooks/Cole (2000).
>
> [2] Zhang, Junbo, Yu Zheng, and Dekang Qi. "Deep spatio-temporal residual networks for citywide crowd flows prediction." AAAI, 2017.
>
> [3] Lin, Ziqian, et al. "Deepstn+: Context-aware spatial-temporal neural network for crowd flow prediction in metropolis." AAAI, 2019.
>
> [4] Hou, Chenyu, et al. "A deep-learning prediction model for imbalanced time series data forecasting." Big Data Mining and Analytics 4.4 (2021): 266-278.
>
> [5] Lee, Hyunwook, et al. "Learning to remember patterns: pattern matching memory networks for traffic forecasting." ICLR, 2022
>
> [6] Lee, Hyunwook, and Sungahn Ko. "TESTAM: a time-enhanced spatio-temporal attention model with mixture of experts." ICLR, 2024
>
> [7] Tang T, Luo W, Huang H, et al. Language-specific neurons: The key to multilingual capabilities in large language models. ACL, 2024
>
> [8] Dai, Damai, et al. "Knowledge neurons in pretrained transformers." ACL (2022).
>
> [9] Voita, Elena, Javier Ferrando, and Christoforos Nalmpantis. "Neurons in large language models: Dead, n-gram, positional."  ACL (Findings) 2024.
>
> [10] Chen, Yuheng, et al. "Journey to the center of the knowledge neurons: Discoveries of language-independent knowledge neurons and degenerate knowledge neurons." AAAI 2024.
>
> [11] Geshkovski, Borjan, et al. "A mathematical perspective on transformers." arXiv preprint arXiv:2312.10794 (2023).

---

> ### Author Response · Authors · 2024-11-26
>
> Dear Reviewer 3uYm,
>
>
> Thank you very much for your thoughtful comments and insightful suggestions. We sincerely appreciate the time and effort you have invested in reviewing our work.
>
>
> We have made our utmost effort to address your comments and have revised our manuscript to incorporate your valuable suggestions. In the revision, we defined low-frequency events (Section 2), conducted additional experiments (Section 4.2), expanded the related work (Section 5), added future directions (Section 6), and expanded the evaluation metric description (Appendix 5). These updates are highlighted in red.
>
>
> We are greatly encouraged by the positive feedback from other reviewers, which has led to increased scores. We kindly ask if you have any additional questions or concerns, as we would be happy to engage in further discussions to address them. If we have successfully addressed most of your concerns, we would greatly appreciate it if you might consider raising your score. Thank you once again for your valuable time and effort.
>
> Best regards,
>
> The Authors

---

### Official Review · Reviewer_XvJ7 · 2024-11-03

**Soundness:** 2
**Presentation:** 3
**Contribution:** 3
**Rating:** 6
**Confidence:** 5

**Summary:**

This work introduces PN-Train, a novel urban time series forecasting method that significantly improves prediction accuracy by identifying and fine-tuning pattern neurons associated with low-frequency events, such as holidays. PN-Train employs a perturbation-based detector to recognize these neurons and enhances them through a fine-tuning mechanism without compromising the representation learning of high-frequency patterns. Empirical results demonstrate that PN-Train considerably boosts forecasting accuracy for low-frequency events while maintaining high performance for high-frequency events.

**Strengths:**

(1) The work proposes a perturbation-based detector that effectively identifies pattern neurons associated with low-frequency patterns.

(2) It designs a fine-tuning mechanism to enhance these pattern neurons without compromising the representation of high-frequency patterns. This design ensures high accuracy for both low-frequency and high-frequency events.

(3) The study provides empirical results that verify the effectiveness of the proposed method, demonstrating significant performance improvements.

**Weaknesses:**

(1) The distinction between low-frequency and high-frequency events is primarily between holidays and non-holidays, which may not have substantial practical significance. Other baselines also improve accuracy for both low-frequency and high-frequency events simultaneously. Focusing on anomaly detection or traffic prediction under anomalous events might have been more meaningful.

(2) The absence of open-source code makes it difficult to reproduce the results.

(3) The study utilizes too few datasets. It would have been better to use datasets from the PeMS system, such as Large-ST [1], which are well-processed and cover longer time spans.

[1] Liu X., Xia Y., Liang Y., et al. LARGEST: A Benchmark Dataset for Large-Scale Traffic Forecasting. Advances in Neural Information Processing Systems, 2024, 36.

**Questions:**

Refer to weaknesses

---

> ### Author Response · Authors · 2024-11-22
>
> We sincerely appreciate your insightful comments. Below, we provide detailed responses to the specific concerns you raised.
>
> > **W1: The distinction between low-frequency and high-frequency events is primarily between holidays and non-holidays, which may not have substantial practical significance. Other baselines also improve accuracy for both low-frequency and high-frequency events simultaneously. Focusing on anomaly detection or traffic prediction under anomalous events might have been more meaningful.**
>
> **[Substantial practical significance]** Accurate forecasting of urban time series during holidays has been recognized as crucial [2, 3, 4, 5], as holiday and non-holiday periods exhibit distinct patterns that can reduce forecasting accuracy during holiday periods. Effectively predicting both patterns is key for urban planning, optimizing schedules, improving operations, and balancing ride-hailing demand and supply by aligning resources with crowd dynamics.
>
> **[Other baselines]** While other baselines improve accuracy for both events, we show that fine-tuning at the neuron level further enhances the network's forecasting ability. For example, although STAEformer achieves strong overall accuracy on the Pedestrian dataset by optimizing for both event types, it tends to prioritize dominant non-holiday patterns, weakening holiday predictions. PN-Train builds on STAEformer by further fine-tuning the pattern neurons, resulting in a 7.91% WMAPE reduction for holidays and a 4.73% reduction for non-holidays compared to STAEformer. These results highlight the value of granular fine-tuning in improving the network's performance on low-frequency patterns.
>
> **[Under anomalous events]** We have conducted additional experiments on forecasting during anomalous events, e.g., extreme weather, with results shown below. The results confirm that fine-tuning at the neuron level significantly improves predictions in these conditions.
>
> **Metro-Traffic**:
>
> |Method||Holiday|||Extreme Weather|||Others|||Overall||
> |:-:|:-:|:-:|:-:|:-:|:--:|:-:|:-:|:-:|:-:|:-:|:-:|:-:|
> ||MAE|RMSE|WMAPE|&nbsp;MAE|RMSE|WMAPE&nbsp;|&nbsp;MAE|RMSE|WMAPE|&nbsp;MAE|RMSE|WMAPE|
> |STAEformer|409.43|721.50|16.19%&nbsp;|&nbsp;1011.52|1511.97|50.45%&nbsp;|&nbsp;214.60|358.94|6.37%&nbsp;|&nbsp;221.37|379.29|6.61%|
> |TESTAM|453.82|734.22|17.94%&nbsp;|&nbsp;1256.48|1791.29|62.67%&nbsp;|&nbsp;337.00|561.62|10.00%&nbsp;|&nbsp;342.19|572.94|10.22%|
> |PN-Train|**406.33**|**718.56**|**16.00%**&nbsp;|&nbsp;**966.82**|**1454.16**|**48.22%**&nbsp;|&nbsp;**213.15**|**357.59**|**6.33%**&nbsp;|&nbsp;**219.77**|**377.33**|**6.57%**|
>
>
>
> **Pedestrian**:
>
> |Method||Holiday|||Extreme Weather|||Others|||Overall||
> |:-:|:-:|:-:|:-:|:-:|:--:|:-:|:-:|:-:|:-:|:-:|:-:|:-:|
> ||MAE|RMSE|WMAPE&nbsp;|&nbsp;MAE|RMSE|WMAPE&nbsp;|&nbsp;MAE|RMSE|WMAPE&nbsp;|&nbsp;MAE|RMSE|WMAPE|
> |STAEformer|120.15|296.71|31.51%&nbsp;|&nbsp;97.76|193.38|30.77%&nbsp;|&nbsp;82.44|203.25|22.66%&nbsp;|&nbsp;84.02|207.19|23.10%|
> |TESTAM|**111.29**|286.88|**29.19%**&nbsp;|&nbsp;116.99|235.39|36.82%&nbsp;|&nbsp;93.59|218.60|25.73%&nbsp;|&nbsp;94.57|221.67|26.00%|
> |PN-Train|115.85|**283.23**|30.39%&nbsp;|&nbsp;**89.89**|**178.33**|**28.29%**&nbsp;|&nbsp;**79.80**|**196.79**|**21.93%**&nbsp;|&nbsp;**81.24**|**200.28**|**22.34%**|

---

> ### Author Response · Authors · 2024-11-22
>
> > **W2: The absence of open-source code makes it difficult to reproduce the results.**
>
> The code, datasets, and checkpoints are available at https://anonymous.4open.science/r/PN-Train/README.md
>
>
> > **W3: The study utilizes too few datasets. It would have been better to use datasets from the PeMS system, such as Large-ST [1], which are well-processed and cover longer time spans.**
>
> We have conducted additional experiments on the Large-ST [1] in the Greater Bay Area, focusing on traffic predictions during holidays and anomalous events like parades. The results below confirm that our training method enhances forecasting for low-frequency events without compromising overall accuracy by neuron-level fine-tuning.
>
> **During holiday**:
>
> |Method||Holiday|||Non-Holiday|||Overall||
> |:-:|:-:|:-:|:-:|:-:|:-:|:-:|:-:|:-:|:-:|
> ||MAE|RMSE|WMAPE&nbsp;&nbsp;|&nbsp;&nbsp; MAE|RMSE|WMAPE&nbsp;&nbsp;|&nbsp;&nbsp;MAE|RMSE|WMAPE|
> |STAEformer|32.39|45.65|8.67%&nbsp;&nbsp;|&nbsp;&nbsp;25.33|39.92|8.20%&nbsp;&nbsp;|&nbsp;&nbsp;25.45|40.01|8.21%|
> |TESTAM|33.86|48.04|9.06%&nbsp;&nbsp;|&nbsp;&nbsp;28.08|43.53|9.09%&nbsp;&nbsp;|&nbsp;&nbsp;28.18|43.60|9.09%|
> |PN-Train|**31.45**|**44.28**|**8.42%**&nbsp;&nbsp;|&nbsp;&nbsp;**25.26**|**39.84**|**8.18%**&nbsp;&nbsp;|&nbsp;&nbsp;**25.36**|**39.92**|**8.18%**|
>
> **During holiday and parades**:
>
> |Method||Holiday|||Parade|||Other|||Overall||
> |:-:|:-:|:-:|:-:|:-:|:-:|:-:|:-:|:-:|:-:|:-:|:-:|:-:|
> ||MAE|RMSE|WMAPE|MAE|RMSE|WMAPE|MAE|RMSE|WMAPE|MAE|RMSE|WMAPE|
> |STAEformer|32.39|45.65|8.67%|27.08|37.20|7.68%|25.33|39.92|8.20%|25.45|40.01|8.21%|
> |TESTAM|33.86|48.04|9.06%|31.98|45.34|8.66%|28.08|43.52|9.09%|28.18|43.60|9.09%|
> |PN-Train|**31.79**|**45.53**|**8.51%**|**25.18**|**34.37**|**7.03%**|**25.19**|**39.85**|**8.16%**|**25.30**|**39.94**|**8.16%**|
>
> ---
>
> We sincerely thank you for your constructive comments. Your insights have been invaluable in improving our manuscript. We hope our responses have addressed your concerns and have been incorporated into the revised version, with changes marked in **RED**. If any questions remain, we would be happy to engage in further discussions.
>
> ---
>
> Reference:
>
> [1] Liu X., Xia Y., Liang Y., et al. LARGEST: A Benchmark Dataset for Large-Scale Traffic Forecasting. Advances in Neural Information Processing Systems, 2024, 36.
>
> [2] Luo, Xianglong, Danyang Li, and Shengrui Zhang. "Traffic flow prediction during the holidays based on DFT and SVR." Journal of Sensors 2019.1 (2019): 6461450.
>
> [3] Cools, Mario, Elke Moons, and Geert Wets. "Investigating effect of holidays on daily traffic counts: time series approach." Transportation research record 2019.1 (2007): 22-31.
>
> [4] McElroy, Tucker S., Brian C. Monsell, and Rebecca J. Hutchinson. "Modeling of holiday effects and seasonality in daily time series." Statistics 1 (2018): 1-27.
>
> [5] Hou, Chenyu, et al. "A deep-learning prediction model for imbalanced time series data forecasting." Big Data Mining and Analytics 4.4 (2021): 266-278.

---

> ### Author Response · Authors · 2024-11-24
>
> Dear Reviewer XvJ7,
>
> Thank you very much for your thoughtful comments and insightful suggestions.
>
> We sincerely appreciate the time and effort you have invested in reviewing our work. We have made our utmost effort to address your comments by clarifying several details, releasing the code, and conducting additional experiments in our rebuttal.
>
> If you have any further questions or concerns, we would be happy to engage in additional discussions. We are truly grateful for your time and comments.
>
> Best regards, The Authors

---

> ### Comment · Reviewer_XvJ7 · 2024-11-26
>
> Thank you to the authors for the rebuttal, which has clarified most of my concerns. The experiments are very well executed. Additionally, I hope that in the subsequent manuscipt revision, more details can be added about anomalous events, such as extreme weather conditions, in the text. This is important in case studies and can also attract the reader's attention. Since further experiments are not mandatory in the discussion phase, consider adding other exceptional events beyond extreme weather (such traffic incident, large gatherings such as concerts) to enhance the highlights of the work. The current content is already quite sufficient; this is just a suggestion that adding more than one would be even better. I will increase my score.

---

> > ### Author Response · Authors · 2024-11-26
> >
> > Thank you very much for your insightful suggestions and for raising the score. We have included additional details about anomalous events in Appendix 7 of the latest revision. Your valuable comments are truly inspiring and have greatly enhanced the quality of our work. We sincerely appreciate your time and effort.

---

### Official Review · Reviewer_x92L · 2024-11-05

**Soundness:** 2
**Presentation:** 3
**Contribution:** 2
**Rating:** 6
**Confidence:** 4

**Summary:**

This paper proposes PN-Train, which detect and fine-tune pattern neurons to improve the forecasting performance on low-frequency patterns.

**Strengths:**

- Improving forecasting performance on low-frequency events is interesting and challenging.
- The idea of pattern neuron is intuitive and reasonable.
- The proposed framework outperforms existing works.

**Weaknesses:**

- The framework requires explicitly defined low-frequency patterns, i.e., holidays, and a dataset with low-frequency patterns is also required (as in Eq(3)). It is unclear whether the framework could handle implicit low-frequency patterns.
- In PNV, it will be better to test whether removing PNs leads to significant performance decreasing on high-frequency patterns, since PNs could be critical for all patterns rather than just low-frequency patterns.
- Minor errors. For example, in Fig.2, PNV seems to deactivate neurons which are not pattern neurons.

**Questions:**

- In line2 Algorithm1, why do author filter out low-frequency samples?
- Can we allocate low-frequency patterns by categorizing neuron activations?
- Is the framework able to handle two or more low-frequency patterns?

---

> ### Author Response · Authors · 2024-11-22
>
> We sincerely appreciate your insightful review and comments. Below, we provide detailed responses to the specific concerns you raised.
>
> > **Q1: In line2 Algorithm1, why do authors filter out low-frequency samples?**
>
> We filter out low-frequency samples because PN-Train needs to fine-tune pattern neurons using a small set of samples related to the target event. By randomly sampling a subset of low-frequency samples from the training dataset, this approach ensures effective fine-tuning without over-training on the same samples. Additionally, this approach also ensures a fair comparison with baseline methods, as all training and fine-tuning samples come from the same training dataset.
>
> > **Q2: Can we allocate low-frequency patterns by categorizing neuron activations?**
>
> Yes, this is achievable. To test the effectiveness of this approach, we introduce PN-Train+, which clusters neuron activation patterns and allocates the cluster with the smallest sample size to low-frequency patterns. Similar to our PN-Train, pattern neurons are identified as those consistently activated under these low-frequency patterns, and the model is fine-tuned using samples from this pattern group.
>
> We compare PN-Train+ with PN-Train* (no fine-tuning) and PN-Train (fine-tuning neurons using explicitly defined low-frequency patterns). Results below show that while PN-Train+ improves forecasting for low-frequency patterns, it underperforms our PN-Train across all pattern types. This is likely due to imprecise pattern categorization from implicit defined patterns, which mixes pattern types and reduces fine-tuning effectiveness.
>
> **Metro-traffic**:
>
> |Method||Holiday|||Non-Holiday|||Overall||
> |:-:|:-:|:-:|:-:|:-:|:-:|:-:|:-:|:-:|:-:|
> ||MAE|RMSE|WMAPE&nbsp;&nbsp;|&nbsp;&nbsp; MAE|RMSE|WMAPE&nbsp;&nbsp;|&nbsp;&nbsp;MAE|RMSE|WMAPE|
> |PN-Train*|460.04|846.75|16.35%&nbsp;&nbsp;|&nbsp;&nbsp;208.84|339.77|6.19%&nbsp;&nbsp;|&nbsp;&nbsp;220.00|379.14|6.58%|
> |PN-Train+|435.48|837.10|15.97%&nbsp;&nbsp;|&nbsp;&nbsp;205.95|335.29|6.10%&nbsp;&nbsp;|&nbsp;&nbsp;216.75|374.30|6.47%|
> |PN-Train|**430.40**|**816.50**|**15.78%**&nbsp;&nbsp;|&nbsp;&nbsp;**203.62**|**332.15**|**6.03%**&nbsp;&nbsp;|**214.29**|**369.46**|**6.40%**|
>
> **Pedestrian**:
>
> |Method||Holiday&nbsp;&nbsp; |||Non-Holiday|||Overall&nbsp;&nbsp;||
> |:-:|:-:|:-:|:-:|:-:|:-:|:-:|:-:|:-:|:-:|
> ||MAE|RMSE|WMAPE&nbsp;&nbsp;|&nbsp;&nbsp; MAE|RMSE|WMAPE&nbsp;&nbsp;|&nbsp;&nbsp;MAE|RMSE|WMAPE|
> |PN-Train*|109.01|259.79|29.31%&nbsp;&nbsp;|&nbsp;&nbsp;78.82|196.39|21.70%&nbsp;&nbsp;|&nbsp;&nbsp;80.45|200.33|22.12%|
> |PN-Train+|107.47|257.30|28.90%&nbsp;&nbsp;|&nbsp;&nbsp;79.46|196.58|21.87%&nbsp;&nbsp;|&nbsp;&nbsp;80.98|200.34|22.26%|
> |PN-Train|**106.11**|**253.86**|**28.54%**&nbsp;&nbsp;|&nbsp;&nbsp;**78.35**|**194.72**|**21.57%**&nbsp;&nbsp;| &nbsp;&nbsp;**79.85**|**198.38**|**21.95%**|
>
> > **W1: The framework requires explicitly defined low-frequency patterns, i.e., holidays, and a dataset with low-frequency patterns is also required (as in Eq(3)). It is unclear whether the framework could handle implicit low-frequency patterns.**
>
> In this study, our primary goal is to investigate whether neurons associated with specific patterns exist within deep networks. By using well-defined low-frequency patterns, such as holidays, as explicit examples, we provide strong evidence supporting their existence.
>
> While the current PN-Train framework does not explicitly handle implicit low-frequency patterns, our findings that fine-tuning explicitly defined patterns also enhances forecasting for other patterns. This suggests that PN-Train improves the model's overall pattern recognition capability, which also benefits the modeling of implicit low-frequency patterns.
>
> Furthermore, our new experiments with PN-Train+ demonstrate that handling implicit low-frequency patterns does not necessarily lead to better forecasting performance. This is because some low-frequency patterns may arise from noise, such as sensor malfunctions, which do not contribute to meaningful improvements. Also, implicit identification can incur misclassification, resulting in suboptimal optimization. In contrast, explicitly defining low-frequency patterns ensures precise optimization for more effective and reliable forecasting performance.

---

> > ### Author Response · Authors · 2024-11-22
> >
> > > **Q3: Is the framework able to handle two or more low-frequency patterns?**
> >
> > The framework can handle multiple low-frequency patterns through sequential fine-tuning. Below, we present forecasting results for two low-frequency events, i.e., holidays and extreme weather [1]. These results demonstrate that fine-tuning pattern neurons allows PN-Train to efficiently handle multiple low-frequency events and beat state-of-the-art baselines in overall performance.
> >
> > **Metro-Traffic**:
> >
> > |Method||Holiday|||Extreme Weather|||Others|||Overall||
> > |:-:|:-:|:-:|:-:|:-:|:--:|:-:|:-:|:-:|:-:|:-:|:-:|:-:|
> > ||MAE|RMSE|WMAPE|&nbsp;MAE|RMSE|WMAPE&nbsp;|&nbsp;MAE|RMSE|WMAPE|&nbsp;MAE|RMSE|WMAPE|
> > |STAEformer|409.43|721.50|16.19%&nbsp;|&nbsp;1011.52|1511.97|50.45%&nbsp;|&nbsp;214.60|358.94|6.37%&nbsp;|&nbsp;221.37|379.29|6.61%|
> > |TESTAM|453.82|734.22|17.94%&nbsp;|&nbsp;1256.48|1791.29|62.67%&nbsp;|&nbsp;337.00|561.62|10.00%&nbsp;|&nbsp;342.19|572.94|10.22%|
> > |PN-Train|**406.33**|**718.56**|**16.00%**&nbsp;|&nbsp;**966.82**|**1454.16**|**48.22%**&nbsp;|&nbsp;**213.15**|**357.59**|**6.33%**&nbsp;|&nbsp;**219.77**|**377.33**|**6.57%**|
> >
> > **Pedestrian**:
> >
> > |Method||Holiday|||Extreme Weather|||Others|||Overall||
> > |:-:|:-:|:-:|:-:|:-:|:--:|:-:|:-:|:-:|:-:|:-:|:-:|:-:|
> > ||MAE|RMSE|WMAPE&nbsp;|&nbsp;MAE|RMSE|WMAPE&nbsp;|&nbsp;MAE|RMSE|WMAPE&nbsp;|&nbsp;MAE|RMSE|WMAPE|
> > |STAEformer|120.15|296.71|31.51%&nbsp;|&nbsp;97.76|193.38|30.77%&nbsp;|&nbsp;82.44|203.25|22.66%&nbsp;|&nbsp;84.02|207.19|23.10%|
> > |TESTAM|**111.29**|286.88|**29.19%**&nbsp;|&nbsp;116.99|235.39|36.82%&nbsp;|&nbsp;93.59|218.60|25.73%&nbsp;|&nbsp;94.57|221.67|26.00%|
> > |PN-Train|115.85|**283.23**|30.39%&nbsp;|&nbsp;**89.89**|**178.33**|**28.29%**&nbsp;|&nbsp;**79.80**|**196.79**|**21.93%**&nbsp;|&nbsp;**81.24**|**200.28**|**22.34%**|
> >
> > [1] https://open-meteo.com/en/docs/historical-weather-api
> >
> > > **W2: In PNV, it will be better to test whether removing PNs leads to significant performance decrease on high-frequency patterns, since PNs could be critical for all patterns rather than just low-frequency patterns.**
> >
> > Yes, you are right. Below, we highlight the relevant results, showing that removing PNs (as in D-PN) leads to a significant decrease in performance on high-frequency patterns, with non-holiday MAE rising by approximately 127% in the Metro-Traffic dataset and 121.92% in the Pedestrian dataset. This is mainly because PNs encompass neurons that capture specific patterns, as well as general time series features critical for all patterns. More results and analysis can be found in lines 281–315.
> >
> >
> > **Metro-Traffic**:
> >
> > |Method||Holiday&nbsp;&nbsp; |||Non-Holiday|||Overall&nbsp;&nbsp;||
> > |:-:|:-:|:-:|:-:|:-:|:-:|:-:|:-:|:-:|:-:|
> > ||MAE|RMSE|WMAPE&nbsp;&nbsp;|&nbsp;&nbsp; MAE|RMSE|WMAPE&nbsp;&nbsp;|&nbsp;&nbsp;MAE|RMSE|WMAPE|
> > |D-PN|663.46|1046.40|24.33%&nbsp;&nbsp;|&nbsp;&nbsp;474.02|586.01|14.04%&nbsp;&nbsp;|&nbsp;&nbsp;482.93|615.44|14.43%|
> > |Original|**446.04**|**846.75**|**16.36%**&nbsp;&nbsp;|&nbsp;&nbsp;**208.84**|**339.77**|**6.19%**&nbsp;&nbsp;|&nbsp;&nbsp;**220.00**|**379.14**|**6.58%**|
> >
> > **Pedestrian**:
> >
> > |Method||Holiday&nbsp;&nbsp; |||Non-Holiday|||Overall&nbsp;&nbsp;||
> > |:-:|:-:|:-:|:-:|:-:|:-:|:-:|:-:|:-:|:-:|
> > ||MAE|RMSE|WMAPE&nbsp;&nbsp;|&nbsp;&nbsp; MAE|RMSE|WMAPE&nbsp;&nbsp;|&nbsp;&nbsp;MAE|RMSE|WMAPE|
> > |D-PN|194.53|370.80|52.31%&nbsp;&nbsp;|&nbsp;&nbsp;174.92|321.45|48.15%&nbsp;&nbsp;|&nbsp;&nbsp;175.98|324.31|48.38%|
> > |Original|**109.01**|**259.79**|**29.31%**&nbsp;&nbsp;|&nbsp;&nbsp;**78.82**|**196.39**|**21.70%**&nbsp;&nbsp;|&nbsp;&nbsp;**80.45**|**200.33**|**22.12%**|
> >
> >
> > > **W3: Minor errors.**
> >
> > Thank you for pointing this out. We have corrected the minor errors in the revised version with changes highlighted in red.
> >
> >
> > ---
> >
> > We sincerely thank you for your constructive comments. Your insights have been invaluable in improving our manuscript. We hope our responses have addressed your concerns. If any queries remain, we would be happy to engage in further discussions and will update the revision accordingly as soon as possible.

---

> ### Author Response · Authors · 2024-11-24
>
> Dear Reviewer x92L,
>
> Thank you very much for your thoughtful comments and insightful suggestions. We sincerely appreciate the time and effort you have invested in reviewing our work.
>
> We have made our utmost effort to address your comments by clarifying several details and conducting additional experiments in our rebuttal.
>
> If you have any further questions or concerns, we would be happy to engage in additional discussions. We are truly grateful for your time and comments.
>
> Best regards, The Authors

---

> > ### Comment · Reviewer_x92L · 2024-11-25
> >
> > Thanks for the authors' detailed response. Most of my concerns have been well addressed.
> > About W1, my concern is that in some real scenarios, we might be unaware about low-frequency patterns, e.g., unrecorded accident, how to make prediction in such scenarios is more interesting and challenging. Still, I appreciate the efforts made by the authors and decide to increase my score to 6.

---

> > > ### Author Response · Authors · 2024-11-25
> > >
> > > Thank you very much for raising the score. Your insightful comments are inspiring and enhance the quality of our work. We greatly appreciate your dedicated time and effort.

---

### Author Response · Authors · 2024-12-03

Dear Reviewers and ACs,

We sincerely thank you for your thoughtful reviews and invaluable comments, which have greatly enhanced the quality of our submission. We are pleased that our additional clarifications and experiments have effectively addressed the concerns raised. Your insightful suggestions have been carefully incorporated into the revised version, with edits temporarily highlighted in **RED**.

We are delighted that the reviewers recognized the **novelty** of PN-Train (Reviewers XvJ7, 3uYm, T7Mg), which **combines neuron-level analysis with practical forecasting tasks** (Reviewer T7Mg) to **enhance accuracy for low-frequency events while maintaining high performance on high-frequency events** (Reviewers T7Mg, XvJ7). By **identifying and fine-tuning neurons associated with low-frequency patterns**, PN-Train **outperforms existing methods** (Reviewer x92L), supported by **solid empirical results** (Reviewers 3uYm, T7Mg) and **visualizations** (Reviewer T7Mg). The paper is also **well-written with sufficient technical details** (Reviewer 3uYm).

The reviewers also raised insightful and constructive concerns, and in response, we have made the following major revisions:
- **Clarifications (Reviewers XvJ7, T7Mg)**: We have emphasized the importance of low-frequency events in urban time series forecasting and clarified the critical role of accurate holiday forecasting in optimizing operations, reducing costs, and enhancing user experiences.

- **Related Work (Reviewers XvJ7, 3uYm, T7Mg)**: We have expanded the related work section to highlight the differences between PN-Train and prior studies on low-frequency events, continuous learning, fine-tuning, and neuron disentanglement.
- **Additional Experiments (Reviewers x92L, XvJ7, T7Mg)**: We have conducted further experiments on various low-frequency events, demonstrating that neuron-level fine-tuning significantly enhances predictions under different anomalous conditions (Appendix 6). We also validated PN-Train on additional datasets (Appendix 6).
- **Baselines (Reviewer 3uYm)**: We have added additional baseline methods to confirm PN-Train’s superior performance in forecasting under low-frequency events (Section 4, Appendix 6).
- **Code Release (Reviewer XvJ7)**: To ensure reproducibility, we have released our code, dataset, and checkpoints.
- **Visualizations (Reviewer T7Mg)**: We have included additional visualizations of pattern neurons to validate our assumptions and highlight our contributions (Appendix 7).

We are greatly encouraged by the positive feedback from all reviewers and delighted that the consensus places our paper above the threshold. We hope PN-Train provides a fresh perspective and inspires further exploration of neuron-level analysis in time series models.

Thank you once again for your invaluable suggestions, support, and organization.

Best regards,

The Authors

---

### Meta-Review · Area_Chair_vaUG · 2024-12-22

**Metareview:**

This paper presents a novel method consisting of a perturbation-based detector and a fine-tuning mechanism for time-series forecasting. The authors introduced neural importance - a novel interpretability mechanism for time-series forecasting task, applicable to both high-frequency and low-frequency events. The experiments on a series of time series forecasting tasks show the superiority of PN-Train, which enhances performance in low-frequency events while maintaining high performance in high-frequency events.
The paper is also well-written.

There were concerns about identifications and details of high and low frequency patterns raised by some reviewers. There were only two datasets used.

The authors have addressed the reviewers' concerns on the details on high and low frequency patterns, and have added experiments on a new dataset, in addition to the two datasets they have in the original version of the paper. These are now added to the appendix of the paper (from LargeST). I strongly suggest that the new experiments to be included in the main body of the paper, in addition to the two datasets, to illustrate the generality of the method.

Although this paper is borderline, given that almost all reviewers have raised their score, and all is unanimous with score 6, so this is still around borderline. In my view, the interpretability of time-series forecasting black box model here is the strongest contribution of this paper that's worth further discussions with others in the conference.

**Additional Comments On Reviewer Discussion:**

There is no further discussions among reviewers during the discussion period. All reviewers are unanimous on score 6. Several reviewers raised their score during author rebuttal.

---

### Decision · Program_Chairs · 2025-01-22

Accept (Poster)